# High Basal Expression and Dual Stress Responsiveness of Soybean (*Glycine max*) Resistance Gene *SRC4*

**DOI:** 10.3390/plants14182820

**Published:** 2025-09-09

**Authors:** Zikai Zhou, Zhuo Bao, Di Miao, Yuxi Zhou, Niu Niu, Hada Wuriyanghan

**Affiliations:** Key Laboratory of Forage and Endemic Crop Biotechnology, Ministry of Education, School of Life Sciences, Inner Mongolia University, Hohhot 010070, China; 67983286@163.com (Z.Z.); nmbzhuo@163.com (Z.B.); 13847685751@163.com (D.M.); zyxl620@163.com (Y.Z.); 13150811843@163.com (N.N.)

**Keywords:** *Glycine max*, *SRC4*, Ca^2+^ signaling, salicylic acid, expression pattern, temperature stress

## Abstract

Genes involved in disease resistance are crucial for plant immune systems, yet their transcriptional regulatory mechanisms remain poorly understood. *SRC4*, a key member of the soybean mosaic virus resistance cluster (SRC), encodes a Ca^2+^-binding EF-hand domain and possesses antiviral activity, but its expression regulation is unclear. Here, we systematically analyzed 4085 soybean (*Glycine max*) transcriptome datasets and conducted SMV inoculation experiments to characterize *SRC4* expression patterns. *Cis*-acting element analysis identified 12 regulatory elements in the *SRC4* promoter, including salicylic acid (SA)-responsive elements. Furthermore, a *Pro_SRC4_*::*GUS* reporter vector was constructed and functional analysis was performed in tobacco (*Nicotiana benthamiana*) and transgenic *Arabidopsis thaliana*. *SRC4* exhibited significantly higher basal expression than typical resistance genes (*R* genes) and was induced by SMV infection, SA treatment, and Ca^2+^ supplementation, with peak expression at 2–5 h post-treatment (hpi). In transgenic tobacco overexpressing *NahG*, neither SMV nor Ca^2+^ could induce *Pro_SRC4_*::*GUS* expression, demonstrating that *SRC4* transcriptional regulation is mediated through SA signaling pathways. *SRC4* showed predominant expression in roots and leaves and responded to temperature stress. Transgenic plants overexpressing *SRC4* exhibited enhanced tolerance to both 12 °C and 37 °C temperature stress. This study elucidates the molecular mechanisms underlying *SRC4* transcriptional regulation through Ca^2+^ and SA signaling pathways, revealing its dual role in both biotic and abiotic stress responses, especially in temperature stress.

## 1. Introduction

Plant diseases constitute one of the primary threats to global food security, causing annual crop yield losses of 20–40% and posing severe challenges to feeding the world’s growing population [1,2]. The intensification of climate change has further exacerbated this challenge through the emergence and expanding geographical distribution of novel pathogenic strains, making the development of sustainable crop disease resistance strategies increasingly urgent [3,4]. Among various disease management approaches, the utilization of resistance genes (*R* genes) for breeding disease-resistant cultivars is widely recognized as the most economical, environmentally sustainable, and effective strategy, circumventing excessive reliance on chemical pesticides and their associated environmental and human health risks [5,6].

Plant *R* genes encode immune receptor proteins capable of recognizing pathogen-associated molecular patterns (PAMPs) or effector molecules, subsequently activating downstream defense responses to combat pathogen invasion [7,8]. Among these, nucleotide binding site-leucine rich repeat (NBS-LRR) genes constitute the largest family of plant *R* genes, accounting for over 70% of cloned *R* genes [9,10]. Proteins encoded by this gene family possess a conserved tripartite domain organization: a variable N-terminal domain (containing TIR or CC domains), a central NBS domain, and a C-terminal LRR domain, enabling recognition of effector molecules from diverse pathogens including bacteria, fungi, viruses, and nematodes [11,12]. Throughout evolutionary processes, plant *R* genes have developed sophisticated expression regulatory systems, with the most notable characteristic being that most *R* genes maintain relatively low basal expression levels under pathogen-free conditions [13,14]. This “low expression-high responsiveness” regulatory pattern is considered an evolutionary strategy allowing plants to achieve balance between defense efficacy and fitness costs. Transcriptomic studies reveal that approximately 72% of NBS-LRR genes in *Arabidopsis thaliana* remain in low expression states under normal conditions, becoming significantly activated only upon pathogen invasion or other stress signal stimulation [15,16].

This stringent expression regulatory strategy primarily stems from the significant fitness costs associated with *R* gene overexpression. Extensive research confirms that constitutive activation of *R* genes often leads to severe growth and developmental defects, including plant dwarfism, leaf necrosis, flowering delays, and yield reductions [17,18]. For example, gain-of-function mutations in the *SNC1* gene in *Arabidopsis* result in constitutive defense response activation accompanied by significant growth inhibition and biomass reduction [19]. Similarly, overexpression of the *Prf1* gene in tomato (*Solanum lycopersicum*) causes constitutive defense activation and growth defects [20]. These fitness costs are more pronounced under field conditions, where plants carrying active *R* genes exhibit up to 10% fitness loss in pathogen-free environments [21].

The evolutionary advantage of inducible expression patterns lies in achieving precise balance between pathogen threats and metabolic costs. When plants perceive PAMPs or effector molecules, *R* genes rapidly upregulate expression, activating multilayered defense mechanisms including reactive oxygen species (ROS) burst, callose deposition, antimicrobial compound synthesis, and hypersensitive responses (HR) [7,22]. This spatiotemporally specific activation strategy ensures that plants can initiate effective defense when facing genuine threats while avoiding unnecessary resource consumption in safe environments [17,23].

Transcriptional regulation of *R* genes involves complex molecular networks encompassing multilevel positive and negative regulatory mechanisms. At the promoter level, *R* genes contain various *cis*-regulatory elements such as pathogen-responsive elements, hormone-responsive elements, and stress-responsive elements, which bind to corresponding transcription factors for precise gene expression regulation [24,25]. Epigenetic modifications also play important roles, with DNA methylation and histone modifications participating in maintaining the basal expression suppression state of *R* genes [26,27]. Environmental factors significantly influence *R* gene expression regulation, reflecting the environmental adaptability of plant immune systems. Almost all environmental changes, whether biotic or abiotic stresses, lead to alterations in *R* gene expression levels, a response pattern distinctly different from other stress-responsive genes [14].

Calcium ions (Ca^2+^) and salicylic acid (SA) serve as early signaling molecules and core defense hormones in plant immune responses, respectively, playing crucial roles in plant disease resistance [28,29,30]. Ca^2+^-SA signal transduction does not operate as two independent pathways but forms a highly integrated signaling cascade through sophisticated spatial molecular networks [31,32]. However, whether these signaling pathways function cooperatively within the same cell types remains poorly understood, and the cell-type-specific interactions between Ca^2+^ and SA signaling represent a critical research direction requiring further investigation.

When plants recognize PAMPs or effector molecules, they rapidly activate plasma membrane and intracellular Ca^2+^ channels, leading to transient elevation of cytoplasmic Ca^2+^ concentrations [33,34]. These Ca^2+^ signals possess specific spatiotemporal patterns that can be precisely recognized and decoded by intracellular Ca^2+^-sensing proteins. In SA biosynthesis regulation, CBP60g serves as a key Ca^2+^-responsive transcription factor, sensing Ca^2+^ signal changes through its conserved calmodulin-binding domain [35,36]. In *sard1 cbp60g* double mutants, pathogen-induced *ICS1* upregulation and SA accumulation are almost completely blocked, resulting in basal resistance defects and loss of systemic acquired resistance (SAR) [31].

The molecular mechanisms of Ca^2+^-SA signaling cascades involve multilayer regulatory networks. Calmodulin-binding transcriptional activator (CAMTA) family proteins serve as important negative regulatory factors, playing key roles in Ca^2+^ signal transduction [32]. CAMTA1, CAMTA2, and CAMTA3 negatively regulate SA biosynthesis by directly suppressing *CBP60g* and *SARD1* gene expression, while pathogen invasion-induced Ca^2+^ signal changes lead to alterations in CAMTA protein activity, thereby relieving suppression of SA biosynthesis genes [37,38]. This dual regulatory mechanism ensures that SA biosynthesis is strictly suppressed under normal conditions while being rapidly activated under pathogen stress. Similarly, abiotic stresses such as drought and salt stress also regulate plant disease resistance capacity by affecting Ca^2+^-SA signal transduction [39].

Recent research has revealed close cooperative relationships between Ca^2+^ signals and NLR-mediated immune responses in immune signal transduction, cell death induction, and inflammatory regulation [40]. NLR proteins can directly function as non-selective cation channels, thereby promoting Ca^2+^ influx into cells [41]. In plant immune systems, ZAR1 resistosomes have been discovered to form Ca^2+^ channels, thereby triggering programmed cell death (PCD) to prevent pathogen spread [42,43]. Additionally, NLRP3 inflammasome activation also depends on Ca^2+^ signal regulation, with Ca^2+^ elevation promoting NLRP3-ASC binding and further triggering inflammatory responses [44].

The effectiveness of plant immune systems depends not only on pathogen invasion but is also significantly regulated by various environmental factors such as temperature, humidity, and light [45]. Among these, temperature, as one of the most critical environmental factors, exerts profound influences on plant–pathogen interaction processes, with regulatory mechanisms primarily manifested in the following aspects.

First, at the transcriptional level, temperature changes can regulate the expression intensity and spatiotemporal patterns of *R* genes through multiple mechanisms [45,46]. Studies demonstrate that many NBS-LRR resistance genes exhibit upregulated expression at the transcriptional level under low-temperature conditions, which may represent an adaptive strategy for plants responding to increased pathogen invasion risks in low-temperature environments [47]. Conversely, high-temperature stress often suppresses the expression of certain *R* genes at the transcriptional level, leading to increased plant susceptibility to pathogens, a phenomenon particularly concerning in the current context of global warming [48].

Second, plant immune system signaling networks exhibit extensive molecular interactions with environmental sensing systems. Temperature changes are often accompanied by fluctuations in intracellular calcium ion concentrations, and Ca^2+^ signals, serving as common mediators for temperature sensing and immune activation, can significantly influence the expression of immune-related genes, potentially representing a key link connecting abiotic and biotic stress responses [45,49]. However, important gaps remain in current research. Studies on transcriptional regulatory mechanisms of most known *R* genes under different temperature conditions are relatively limited, particularly the molecular mechanisms of how high basal expression *R* genes respond to temperature changes at the transcriptional level remain insufficiently elucidated [50]. This research deficiency is particularly prominent in the context of climate change. For example, rice (*Oryza sativa*) has specific temperature requirements at different developmental stages: the optimal temperature for seed germination is 28–32 °C, while the optimal temperature for heading stage is 25–35 °C, with supra-optimal temperature stress leading to significant losses in yield and quality [51,52]. In the current environment of frequent extreme temperature events, plants face compound pressures from both biotic and abiotic stresses, requiring plant immune systems to possess stronger environmental adaptability and signal integration capabilities.

Therefore, in-depth analysis of how *R* genes respond to temperature and other environmental factors at the transcriptional level holds important theoretical and practical significance for breeding climate-adaptive disease-resistant varieties. This not only contributes to our deeper understanding of the environmental adaptability of plant immune systems but also provides scientific foundation for sustainable agricultural development under the challenges of climate change.

In earlier work, we reported a soybean mosaic virus (SMV) resistance gene cluster SRC (SMV resistance cluster) in soybean (*Glycine max*) cultivar Williams, located on chromosome 16 and containing tandemly arranged NBS-LRR genes (*SRC1-SRC13*) [53]. Functional validation demonstrates that the SRC gene cluster confers broad-spectrum resistance to multiple SMV strains in soybean. Among these, the *SRC4* gene exhibits unique functional characteristics: unlike other SRC genes, *SRC4* not only participates in SMV resistance but also encodes a Ca^2+^-binding EF-hand domain. Genome-wide association studies (GWAS) of soybean powdery mildew disease (PMD) also confirmed that *SRC4* shows strong association with PMD resistance [54]. More importantly, *SRC4* maintains relatively high basal expression levels without affecting normal plant growth and development, suggesting it may possess unique transcriptional regulatory mechanisms and biological functions.

Based on this research background, this study aims to address key questions surrounding *SRC4*: Does *SRC4* truly possess high basal expression characteristics that distinguish it from typical *R* genes? What are the special features of its transcriptional regulatory patterns? Can *SRC4* respond to abiotic stresses such as temperature? What is the tissue-specific expression pattern of *SRC4*? How does this expression characteristics affect plant growth, development, and stress tolerance? Through integrating multiple technical approaches including transcriptomic analysis, promoter functional validation, signal transduction analysis, and transgenic functional studies, this research is expected to provide new theoretical perspectives for understanding the diversity of *R* gene expression regulation, supply important genetic resources and theoretical foundations for breeding soybean varieties with broad-spectrum resistance and environmental adaptability under climate change contexts, and contribute new scientific insights to the study of co-evolutionary mechanisms between plant immune systems and environmental adaptability.

## 2. Results

### 2.1. Expression Pattern of SRC4

Previously, we reported that the *SRC4* gene in the SRC gene cluster exhibits antiviral activity [53]. To further investigate the expression characteristics of *SRC4* during soybean growth, we systematically analyzed its expression patterns using the PlantRNAdb (Plant Public RNA-seq Database) transcriptome database (https://plantrnadb.com/, accessed on 15 November 2023). We analyzed 4085 soybean transcriptome datasets, including various treatment conditions, and the results showed that *SRC4* exhibited high basal expression levels across all transcriptome samples. As shown in Figure 1A, *SRC4* FPKM (fragments per kilobase of transcript per million mapped reads) values were mainly distributed in the 0–20 range. This expression level was significantly higher than the typical expression patterns of other *R* genes. To validate this observation, we also examined the expression levels of several other well-characterized soybean *R* genes, including soybean mosaic virus (SMV) *R* genes such as the *Rsv1* candidate gene *3gG2*, the *RSMV-11* candidate gene *GmMATE68*, and *Rsv3*, as well as the *Phytophthora sojae R* genes *RpsUN2*. Consistent with typical *R* gene expression patterns, all of these genes, along with the *R* genes *SRC7* reported in our previous work, showed extremely low basal expression levels in the same transcriptome datasets, with FPKM values generally being below 2. This comparative analysis further highlighted the unique high-expression characteristic of *SRC4* among soybean *R* genes.

To validate the results from transcriptome data analysis, we further compared the expression pattern of *SRC4* and *SRC7* through promoter activity analysis. First, we obtained *SRC4* promoter sequence information from the soybean genome database Phytozome (ID: 275, Organism: *Glycine max* Wm82.a2.v1) and cloned the *SRC4* promoter sequence (*Pro_SRC4_*) from the soybean Williams cultivar genome. Subsequently, we used Nimble Cloning (NC) cloning technology to clone the promoter sequence to the binary expression vector pNC-121-Pro to obtain the *Pro_SRC4_*::*GUS* expression vector, while the *Pro_SRC7_*::*GUS* vector was reported in the previous work [53]. After these promoter-*GUS* vectors were transiently expressed in tobacco (*Nicotiana benthamiana*), *GUS* enzyme activity assay was performed. As shown in Figure 1B, compared to *Pro_SRC7_*, *Pro_SRC4_* exhibited much stronger *GUS* activity. At the mRNA level, *Pro_SRC4_*::*GUS* expression was also nine times higher than that of *Pro_SRC7_* (Figure 1C). We further constructed *Pro_SRC4_* and *Pro_SRC7_* into the binary expression vector pNC-Green-LUC to obtain *Pro_SRC4_*::*LUC* and *Pro_SRC7_*::*LUC* vectors, and performed luciferase assay LCA for reporter gene. As shown in Figure 1D, *Pro_SRC4_* induced stronger bioluminescence signals as comparison to *Pro_SRC4_*. At the mRNA level, *Pro_SRC4_*::*LUC* also showed 6.5 times higher accumulation than that by *Pro_SRC7_*::*LUC* (Figure 1E).

### 2.2. Transcriptional Regulatory Analysis of SRC4

Given that *SRC4* exhibited high basal expression levels in soybean, which contrasts sharply with the low expression patterns of typical disease *R* genes, this suggests that *SRC4* may possess unique transcriptional regulatory mechanisms. To elucidate the expression regulatory characteristics of *SRC4* gene, we first used the PlantCARE database (http://bioinformatics.psb.ugent.be/webtools/plantcare/html/, accessed on 25 December 2023) to predict *cis*-acting elements in the 2000 bp promoter region (*Pro_SRC4_*) upstream of its transcription start site. The results showed that this region contains 12 different types of *cis*-acting elements, including one salicylic acid (SA)-responsive elements in −1761~−1731 bp region (Figure 2A, Table 1).

SA serves as an important defense hormone in plants, playing a key role in soybean immune responses against SMV. Based on the discovery of SA-responsive elements in the promoter analysis, we treated V3 stage soybean Williams seedlings with a foliar spray of 0.5 mM SA solution. Quantitative RT-qPCR analysis showed that *SRC4* expression was significantly upregulated within 1–5 h after SA treatment, followed by gradual decline (Figure 2B). This result confirms *SRC4*’s responsiveness to SA signals.

Considering the important role of SA in plant antiviral responses, we further analyzed the expression patterns of *SRC4* under viral infection conditions. By mining transcriptome data from three soybean cultivars in the NCBI database—Williams (SRR5466715–SRR5466722), L29 (SRR10098755–SRR10098764), and 5031 (SRR6376246–SRR6376249)—under SMV infection for local analysis, we found that *SRC4* showed significant upregulation in all cultivars upon SMV infection (Figure 2C).

To precisely capture the dynamic expression changes of *SRC4* during early SMV infection, we conducted time-course sampling at 1, 2, 5, 12, and 24 h post-infection, using SMV-N1 strain rub-inoculation on V3 stage soybean leaves. The results showed that *SRC4* responded rapidly at transcription level after viral infection, with peak expression occurring at 2–5 h, exhibiting typical early response gene expression patterns (Figure 2D).

### 2.3. Ca^2+^ Signal-Dependent Regulatory Mechanism of SRC4 Expression

SRC4 protein possesses Ca^2+^-binding EF-hand domain [53]. Tobacco mosaic virus (TMV) infection of tobacco rapidly activates plasma membrane Ca^2+^ channels leading to increased cytoplasmic Ca^2+^ concentrations [55,56], and potato virus X (PVX) infection similarly triggers Ca^2+^ signal transduction [57]. Based on above observations, we hypothesized that transcriptional regulation of *SRC4* may also be regulated by Ca^2+^ signals.

To test this assumption, we applied exogenous Ca^2+^ treatment to evaluate its effect on *SRC4* expression. Three-week-old Williams soybean seedlings were treated with foliar spray of 10 mM CaCl_2_ solution, with time-course sampling at 1, 2, 5, 12, and 24 hpi. RT-qPCR analysis showed that exogenous Ca^2+^ treatment could significantly induce *SRC4* expression, with peak expression occurring at 1–2 hpi by 4.1 folds of the control (Figure 3A). This result indicates that *SRC4* transcriptional levels are indeed subject to positive regulation by Ca^2+^ signals.

To systematically verify the responsiveness of the *SRC4* promoter to multiple signaling molecules, we generated a *Pro_SRC4_*::*GUS* reporter gene construct in the tobacco transient expression system. After transforming *Pro_SRC4_*::*GUS* fusion vectors into tobacco leaves, the following treatments were applied: (1) SMV-GFP infection (OD_600_ = 1.0) to simulate viral stress; (2) 0.5 mM SA spray to simulate defense hormone signals; and (3) 10 mM CaCl_2_ spray to simulate Ca^2+^ signals. *GUS* staining detection was performed 72 hpi. The results showed that all three treatments significantly enhanced *GUS* signal intensity, compared to the basal expression activity of the Mock control group (Figure 3B). RT-qPCR further confirmed that all three treatments significantly upregulated *SRC4* expression at the transcriptional level (Figure 3C), confirming the multiple responsiveness of the *Pro_SRC4_* promoter to viral infection, SA, and Ca^2+^ signals.

Extensive research indicates that Ca^2+^ influx triggered by plant viral infection is a key early event in activating downstream defense responses. Ca^2+^ signals further activate Ca^2+^/calmodulin-dependent protein kinases, phosphorylating and activating key enzymes in SA biosynthesis such as phenylalanine ammonia-lyase (PAL) and isochorismate synthase (ICS), thereby promoting massive SA accumulation. Subsequently, SA, as an important defense signaling molecule, induces expression of numerous disease resistance-related genes including pathogenesis-related (*PR*) genes. Based on the theoretical framework of Ca^2+^-SA signaling cascade regulation, we hypothesized that Ca^2+^ might regulate *SRC4* transcription through modulating SA accumulation. To test this hypothesis, we conducted functional validation experiments using transgenic tobacco overexpressing the *NahG* gene. *NahG* encodes salicylate hydroxylase from *Pseudomonas putida*, which can specifically catalyze the conversion of SA to catechol, thereby blocking SA accumulation in plants, making it a classic tool for studying SA function.

In Ox-*NahG* transgenic tobacco, *Pro_SRC4_*::*GUS* fusion vectors were transiently expressed, and the same SMV infection, SA treatment, and Ca^2+^ treatment were applied. *GUS* staining results showed that under conditions where SA accumulation was blocked, all three treatments failed to induce reporter *GUS* expression, with expression levels showing no significant difference from the Mock control group (Figure 3D). This result provides strong evidence that the inductive effects of SMV infection, SA treatment, and Ca^2+^ treatment on *SRC4* expression all depend on SA accumulation.

### 2.4. Tissue-Specific Expression Pattern of SRC4

To comprehensively analyze the tissue-specific expression pattern of the *SRC4* gene, we systematically analyzed tissue distribution information from 4085 soybean transcriptome datasets in the PlantRNAdb database. Through statistical analysis of expression levels in six major tissue organs, such as roots, leaves, seeds, flowers, embryos, and endosperm, we found that *SRC4* exhibited obvious tissue expression preferences (Figure 4A). *SRC4* showed the highest expression levels in roots, with median FPKM values reaching 20–25, suggesting that this gene may play important functions in root development, nutrient absorption, or root-specific defense responses. Expression levels in leaf tissues ranked second, which is consistent with its functional characteristics as an *R* gene primarily exerting defense functions in above-ground parts. Expression levels in seeds, flowers, and embryo tissues were relatively low, with FPKM values mainly distributed in the 0–15 range. In contrast, *SRC4* mRNA was barely detectable in endosperm, with FPKM values close to 0, indicating extremely low transcriptional activity of this gene in seed storage tissues.

To validate transcriptome data analysis, we further performed RT-qPCR detection on four tissues—roots, leaves, seeds, and flowers—of soybean Williams cultivar. The results were highly consistent with database analysis, confirming that *SRC4* is mainly highly expressed in roots and leaves, while expression levels in seeds and flowers are relatively low (Figure 4B).

To verify the functional characteristics of the *Pro_SRC4_* promoter and the conservation of its tissue expression patterns in model plants, we constructed *Pro_SRC4_*::*GUS* transgenic *Arabidopsis* lines. Twenty-five independent transgenic lines were obtained through floral dip transformation, and after kanamycin resistance screening and genomic PCR verification, positive lines were selected and propagated to T_3_ generation for subsequent analysis. *GUS* histochemical staining of flowering T_3_ transgenic plants showed that *GUS* activity was mainly distributed in leaves, stems, roots, and flowers, which was basically consistent with expression patterns in soybean (Figure 4C), demonstrating the cross-species conservation of *Pro_SRC4_* promoter function.

Furthermore, we verified the responsiveness of *Pro_SRC4_* to SA and Ca^2+^ signals in transgenic *Arabidopsis*. T_3_ transgenic plants were treated with a foliar spray of 0.5 mM salicylic acid (SA) and 10 mM CaCl_2_, respectively, and *GUS* transcriptional levels were detected by RT-qPCR at 48 hpi. The results showed that both SA and Ca^2+^ treatments significantly upregulated *GUS* mRNA expression levels (Figure 4D). Meanwhile, we used quantitative *GUS* protein activity detection to further verify protein-level changes. Total leaf proteins were extracted 72 hpi, protein concentrations were determined by the Bradford method and adjusted to uniform concentration (10 μg/mL), and protein abundance was detected using *GUS* quantitative detection kit on a FilterMax F5 microplate reader (Molecular Devices, San Jose, CA, USA). The results showed that *GUS* protein accumulation in both SA and Ca^2+^ treatment groups was significantly higher than the Mock control group (Figure 4E). To comprehensively characterize the tissue-specific expression pattern of the *SRC4* promoter, we performed RT-qPCR analysis of *GUS* expression across ten distinct tissues from T_3_ *Pro_SRC4_::GUS* transgenic *Arabidopsis* plants These tissues included seedling roots, root tips, cotyledons, rosette leaves, cauline leaves, inflorescence stem, flower buds, open flowers, pistils, and young siliques. The results revealed differential *GUS* expression levels among various tissues, with the highest expression observed in root tissues (seedling roots and root tips), moderate expression in vegetative organs (cotyledons, rosette leaves, and cauline leaves), and relatively lower expression in reproductive tissues (flower buds, open flowers, pistils, and young siliques) and inflorescence stem (Figure 4F).

These results indicate that *SRC4* has clear tissue expression preferences, primarily functioning in roots and leaves, and its promoter maintains responsiveness to SA and Ca^2+^ signals in heterologous systems.

### 2.5. Functional Analysis of SRC4 Response to Abiotic Stress

Based on previous studies of *SRC4* expression characteristics under biotic stress conditions, we further explored the expression patterns of this gene under abiotic stress. Through analysis of soybean transcriptome data in the PlantRNAdb database, we found that *SRC4* not only showed induced expression under pathogen stress but also exhibited significant expression upregulation under low-temperature and high-temperature stress conditions (Figure 5A). Compared to normal conditions, both cold stress and heat stress treatments significantly increased *SRC4* transcriptional levels.

To verify the responsiveness of the *Pro_SRC4_* promoter to temperature stress, we conducted functional validation in the tobacco transient expression system. After transforming *Pro_SRC4_*::*GUS* reporter vectors into tobacco leaves, 12 °C low-temperature treatment and 37 °C high-temperature treatment were applied, respectively, and *GUS* staining detection was performed 7 days post-treatment. The results showed that both low-temperature and high-temperature stress significantly induced enhanced *GUS* activity, compared to the 22 °C control group (Figure 5B), confirming the responsiveness of the *Pro_SRC4_* promoter to temperature changes.

Furthermore, we validated the above findings via the analysis in *Pro_SRC4_*::*GUS* transgenic *Arabidopsis*. T_3_ transgenic plants were subjected to the same temperature stress treatments, and *GUS* mRNA levels were detected by RT-qPCR at 25 days post-treatment. The results showed that both 12 °C and 37 °C treatments significantly upregulated *GUS* expression, which was highly consistent with tobacco transient expression results (Figure 5C).

To evaluate the effect of *SRC4* overexpression on plant temperature stress tolerance, we constructed transgenic *Arabidopsis* lines overexpressing *SRC4*. The full-length *SRC4* sequence was cloned into the pNC-Cam3304-MCS35S vector. Through herbicide resistance screening, genomic PCR verification, and T_2_ generation RT-qPCR analysis, we successfully obtained Ox-*SRC4* transgenic *Arabidopsis* lines stably expressing *SRC4*, which were propagated to T_3_ generation for phenotypic analysis (Appendix A).

In temperature stress tolerance tests, T_3_ Ox-*SRC4* transgenic plants and wild-type Col-0 simultaneously received 12 °C low-temperature stress and 37 °C high-temperature stress treatments for 25 days, and plant growth status was observed. The results showed that Ox-*SRC4* transgenic lines exhibited significantly better growth performance than wild-type under both temperature stress conditions. Under low-temperature conditions, Ox-*SRC4* transgenic plants maintained normal leaf expansion and growth status; under high-temperature conditions, Ox-*SRC4* transgenic plants also showed stronger tolerance, with significantly reduced leaf wilting (Figure 5D). Similar results were also validated in Ox-*SRC4* transgenic tobacco, which shows higher tolerance to both low and high-temperature treatment compared with 17Wt tobacco (Figure 5E). Notably, the growth of Ox-*SRC4* transgenic tobacco was not inhibited under normal conditions or Ca^2+^ treatment, demonstrating the absence of fitness penalties typically associated with resistance gene overexpression (Appendix A).

These results indicate that the SRC4 gene not only participates in plant biotic stress responses but also plays important roles in abiotic stress adaptation, significantly enhancing plant tolerance to temperature stress.

## 3. Discussion

High expression of *R* genes can enhance plant tolerance to biotic stresses (such as pathogen invasion) or abiotic stresses (such as drought, high salinity, and low temperature). However, prolonged overexpression of such genes often imposes metabolic burdens on plant growth and development, thereby affecting plant growth rates or biomass accumulation. Therefore, to maintain normal growth and development, typical *R* genes usually remain in low expression states under non-stressed conditions, and they are induced and activated only under external environmental stimuli to exert their resistance functions.

Our research represents the first discovery that the *SRC4* gene exhibits a high basal expression pattern significantly different from typical NBS-LRR genes, challenging the classical “low expression-high responsiveness” regulatory paradigm of plant *R* genes. Approximately 72% of NBS-LRR genes maintain constitutive expression at low levels, with most *R* genes exhibiting strictly controlled basal expression to minimize fitness costs [9,15]. Classical *R* genes such as tobacco *N* gene, *Arabidopsis RPS2* and *RPM1* maintain low expression or transcriptionally silenced states under non-stressed conditions [58,59]. Through systematic analysis of 4085 soybean transcriptome datasets, we found that *SRC4* FPKM values were significantly higher than those of the typical *R* gene *SRC7* (FPKM < 2) in the same gene cluster. This fundamental difference in expression patterns suggests that *SRC4* may represent a novel class of *R* genes with unique regulatory mechanisms and biological functions.

Traditional perspectives hold that constitutive expression of *R* genes leads to severe fitness costs, including negative effects such as plant dwarfism, biomass reduction, and the modification of hormonal balance, with subsequent epigenetic re-programming and constitutive activation of defense responses [17,21]. Overexpression of the *R* gene *Prf1* causes constitutive activation of defense responses that severely affect normal tomato growth [20]. Under field conditions, this can cause fitness losses of up to 10% in plant yield. However, our previous research indicated that the high basal expression of *SRC4* does not cause obvious growth and developmental defects, a phenomenon that may be closely related to its unique functional mechanisms. An important explanation is that *SRC4* protein can alter its disease resistance function through binding with Ca^2+^, and this ligand-dependent activation mechanism may enable plants to avoid constitutive activation of defense responses while maintaining high protein abundance. Similar mechanisms have been reported in some calmodulin-dependent kinases, which remain inactive without Ca^2+^ binding and function only under specific signal stimulation [60]. In rice, there is a mechanism through competitive inhibition of *PigmR* homodimerization by *PigmS*, cleverly achieving fine regulation of disease resistance intensity while avoiding adverse effects of excessive immune responses on growth [61].

The high expression of *SRC4* reflects adaptive evolutionary strategies of plants when facing complex and variable environments. Unlike inducible expression that requires *de novo* protein synthesis, high expression ensures rapid availability of defense proteins, enabling plants to quickly initiate defense responses in the very early stages of pathogen invasion. This “pre-storage” strategy is particularly important in viral infections, as viral replication is extremely rapid and traditional transcriptional induction may not be timely enough to prevent early viral establishment [62]. Additionally, we observed high expression of *SRC4* in roots and leaves, suggesting it may possess multiple biological roles beyond disease resistance functions, and this functional diversity may be an important reason for maintaining its high expression.

The expression pattern of *SRC4* may represent adaptive innovation of plant immune systems in specific ecological niches. In environments with high incidence of RNA viruses such as SMV, maintaining high levels of disease resistance proteins may confer greater selective advantages than conserving metabolic costs. In natural populations with high pathogen pressure, certain *R* genes tend to maintain higher basal expression levels [63]. Meanwhile, the high expression pattern of *SRC4* is also consistent with plant “priming” defense strategies, which balance defense efficacy with metabolic costs by maintaining moderate defensive preparedness [64].

Salicylic acid (SA), as a key signaling molecule in plant defense responses, plays a central regulatory role in inducing *R* gene expression [29]. Extensive research confirms that SA can significantly induce expression of pathogenesis-related genes, which are widely used as molecular markers of SAR [65]. In *Arabidopsis* and tobacco, SA treatment can rapidly activate expression of *PR-1*, *PR-2*, and *PR-5* genes, with *PR-1* gene expression being considered the primary marker of SAR establishment [66]. In rice, the *R* gene *OsNPR3.3* was upregulated 4-fold at 24 h after 2 mM SA treatment [67], which is consistent with our findings of SA-induced upregulation of *SRC4* expression. Salicylic acid also functions as a multifunctional endogenous substance for treating plant diseases. It plays important roles in alleviating abiotic stresses including high temperature, low temperature, drought, UV radiation, heavy metals, and osmotic shock [68,69].

Upregulation of *R* gene expression during pathogen invasion is an important phenomenon in plant immune responses. Many NBS-LRR resistance genes exhibit low-level sustained expression in healthy, unchallenged tissues but are upregulated under bacterial flagellin stimulation, consistent with basal resistance induction, indicating the need for rapid responses to pathogen attacks [9]. In maize (*Zea mays*), the *ZmNBS25* gene shows relatively low expression levels in uninfected maize leaves but is significantly upregulated 24 h after inoculation with *Bipolaris maydis* [70]. From an evolutionary perspective, inducible expression may be an adaptive strategy for plants facing pathogen pressure, providing effective defense during pathogen attacks while avoiding fitness costs that might result from overexpression of *R* genes under normal conditions [21]. In our study, *SRC4* similarly showed upregulated expression after SMV infection, which is highly consistent with the expression patterns of the above *R* genes.

As an important second messenger in cells, Ca^2+^ can also induce gene expression upregulation. Early research by Braam (1992) showed that elevated extracellular Ca^2+^ increased expression of genes encoding calcium sensors, and heat shock or cold shock induction of some genes depends on exogenous Ca^2+^ signals [71]. *SRC4* mRNA levels were upregulated by Ca^2+^ addition. Through time-course expression analysis, we found that Ca^2+^ treatment-induced peak *SRC4* expression occurred at 1–2 h, while SMV infection-induced expression peaks occurred at 2–5 h. This temporal difference is consistent with the sequence of Ca^2+^-SA signaling. Ca^2+^, as an early signaling molecule in plant immune responses, can rapidly activate downstream transcriptional regulatory networks, while SA biosynthesis and accumulation as a defense hormone requires longer time [31,32].

The Ca^2+^ signaling pathway is a key regulatory mechanism for initiating salicylic acid biosynthesis, with transcriptional regulation of the *ICS1* gene playing a decisive role [72]. When plant cells recognize pathogens, plasma membrane Ca^2+^ channels rapidly open, causing transient elevation of cytoplasmic Ca^2+^ concentrations. This Ca^2+^ signal subsequently coordinately activates *ICS1* gene transcriptional expression through two functionally complementary parallel pathways—the calmodulin-dependent CBP60g pathway and the calmodulin-independent SARD1 pathway—thereby initiating the salicylic acid biosynthesis process. Notably, CBP60g family transcription factors exhibit high evolutionary conservation across plant lineages, suggesting that the Ca^2+^-SA regulatory network may have ancient evolutionary origins, reflecting the fundamental status of this signaling pathway in plant immune systems [31].

To verify whether the *SRC4* gene is similarly regulated by SA signals, we conducted functional validation experiments using the *NahG* transgenic tobacco system. *NahG*-encoded salicylate hydroxylase can specifically degrade SA, thereby blocking SA accumulation. Experimental results clearly showed that under conditions where SA accumulation was effectively blocked by *NahG* enzyme, Ca^2+^ treatment, exogenous SA treatment, and SMV viral infection all completely lost their inductive effects on *SRC4* gene expression. This key finding strongly demonstrates that SA plays an irreplaceable central role in *SRC4* transcriptional activation and establishes *SRC4*’s molecular identity as a downstream response gene in the SA signaling pathway. This strict SA-dependent regulatory pattern is highly similar to the classical CBP60g-SARD1-ICS1 signaling [35,73], further supporting the hypothesis that *SRC4* participates in plant SA-mediated immune responses.

Analysis of the *SRC4* promoter region provides important molecular evidence for the above regulatory mechanisms. Within the 2000 bp region upstream of the transcription start site, we identified 12 different types of *cis*-acting elements, among which the presence of SA-responsive elements (located at −1761~−1731 bp) directly explains *SRC4*’s strong transcriptional response to SA treatment. Plant SA-responsive elements typically contain TGACG or related sequence motifs that can be recognized and bound by NPR1-TGA transcription factor complexes [74]. Compared to classical SA-responsive genes, the upstream region of SA-responsive elements in the *SRC4* promoter also contains multiple other regulatory elements, including two MYB binding sites (−1144~−1114 bp and −537~−507 bp), two defense and stress-responsive elements (−1591~−1561 bp and −188~−158 bp), and various hormone-responsive elements. This combinatorial element arrangement enables *SRC4* to integrate multiple environmental signals. The dual distribution of defense and stress-responsive elements in the distal and proximal regions of the *SRC4* promoter also represents its unique expression pattern. Distal elements typically participate in maintaining basal expression levels, while proximal elements are more involved in rapid activation of inducible expression [75]. This spatial distribution pattern highly matches our observed *SRC4* expression characteristics of high basal expression with strong inducibility. Additionally, the presence of ABA-responsive and GA-responsive elements suggests that *SRC4* expression is closely related to plant stress states and growth-developmental states, and this integration of growth-defense signals may be an important mechanism for *SRC4* to avoid severe fitness costs [76]. Based on these findings, we speculate that transcriptional regulation of *SRC4* involves complex signal integration networks. Ca^2+^ signals may phosphorylate related transcription factors by activating calcium-dependent protein kinases (CDPKs), while accumulated SA directly activates SA-responsive elements in the *SRC4* promoter through NPR1-TGA complexes. Meanwhile, MYB transcription factors may participate in coordinating light signals with immune signals, while various hormone-responsive elements ensure adaptive expression of *SRC4* under different physiological states.

Our research demonstrates that *SRC4* possesses the unique ability to simultaneously respond to both biotic and abiotic stresses. Through systematic analysis of soybean transcriptome data, we found that *SRC4* not only shows significantly upregulated expression under SMV infection conditions but also exhibits significant transcriptional activation under low-temperature and high-temperature stress conditions. This multiple stress response capability is extremely rare among *R* genes. The regulatory mechanisms of temperature stress on *SRC4* expression may involve molecular networks at multiple levels. The various stress-responsive elements we discovered in the *SRC4* promoter provide a molecular basis for its temperature sensitivity, while Ca^2+^ signals, serving as common mediators for temperature sensing and immune activation, may be the key link connecting these two stress responses. Temperature changes, particularly low-temperature stress, can rapidly activate mechanosensitive Ca^2+^ channels on plant cell membranes, leading to elevated cytoplasmic Ca^2+^ concentrations [77,78]. The spatiotemporal patterns of such Ca^2+^ signal generation may differ from Ca^2+^ influx caused by pathogen invasion, but both can be recognized and responded to by *SRC4*’s regulatory network, thus achieving unified regulation of different types of stress.

More importantly, we confirmed through overexpression plant Ox-*SRC4* experiments that *SRC4*’s multiple stress response capability has important biological functions. Ox-*SRC4* transgenic *Arabidopsis* and tobacco both exhibited significantly better growth performance and stress tolerance than wild-type under 12 °C low-temperature stress and 37 °C high-temperature stress conditions. This result indicates that *SRC4* can not only sense temperature changes but also enhance plant temperature adaptability through the functional activities of its protein products. Considering that *SRC4* protein possesses Ca^2+^-binding capability, we speculate that its temperature protection function may be related to maintaining Ca^2+^ homeostasis. Under temperature stress conditions, drastic fluctuations in intracellular Ca^2+^ concentrations often lead to cellular dysfunction [79], and *SRC4* protein may buffer these fluctuations through its Ca^2+^-binding capability and Ca^2+^ regulatory capacity, thereby protecting cells from temperature stress damage.

The interaction between biotic and abiotic stress signaling pathways is an important strategy for plants to adapt to complex environments. Plants in natural environments often face challenges from multiple stresses simultaneously, and complex interactions exist among these stresses [80]. For example, temperature stress can affect plant susceptibility to pathogens, while pathogen invasion can also alter plant temperature tolerance. *SRC4*’s multiple stress response capability may enable plants to maintain good adaptability in such complex stress networks. When plants simultaneously face viral infection and temperature stress, *SRC4* activation can provide both disease resistance protection and enhanced temperature tolerance, thereby maximizing plant survival opportunities.

From an evolutionary biology perspective, *SRC4*’s multiple stress response characteristics may reflect adaptive evolution of plant immune systems under climate change contexts. With global warming and increasing extreme weather events, plants increasingly need the capability to simultaneously cope with multiple stresses. Traditional single-function *R* genes may face selective disadvantages in such environments [81], while genes with multifunctional characteristics like *SRC4* are more likely to be retained and optimized by natural selection, as maintaining high levels of defense proteins may confer greater selective advantages than conserving metabolic costs in multiple stress environments [82].

Compared to other reported multiple stress response genes, *SRC4* possesses unique functional characteristics. For example, the DREB transcription factor family in *Arabidopsis* can simultaneously regulate drought and cold stress responses but primarily functions through transcriptional regulation [83]. As an immune receptor protein, *SRC4*’s multiple stress response function is more reflected in direct actions at the protein level. This difference suggests that plants may achieve adaptation to complex environments through regulatory mechanisms at different levels, and *SRC4* represents an important example of functional diversification of immune receptor proteins.

Our research reveals that *SRC4* has obvious tissue-specific expression patterns, with its high expression levels in roots and leaves suggesting that this gene may possess multiple biological roles beyond traditional disease resistance functions. Most NBS-LRR resistance genes, such as *Arabidopsis RPM1* and *RPS2*, are mainly expressed in leaves with relatively low expression levels in roots [84]. We found that *SRC4* exhibits the highest expression levels in roots, with leaf tissue expression ranking second, while expression in reproductive tissues such as seeds, flowers, and embryos is relatively low, with almost no expression in endosperm, suggesting that *SRC4* may play important physiological functions in plant vegetative organs. Root systems, as the primary contact interface between plants and soil environments, not only undertake functions of water and nutrient absorption but also face continuous threats from soil-borne pathogens [85]. The rhizosphere environment contains numerous pathogens including bacteria, fungi, and nematodes, which differ significantly from foliar pathogens in invasion strategies and effector molecule composition [86,87]. High expression of *SRC4* in roots may enable plants to rapidly recognize and respond to rhizosphere pathogen invasion, providing first-line defense protection. Reports have also shown that *SRC4* expression is significantly increased after infection with soybean cyst nematode (SCN), indicating that *SRC4* functions in SCN resistance [88]. Additionally, root systems are important organs for plants to perceive environmental changes, and changes in soil temperature, humidity, and ion concentrations are first perceived by root systems. High expression of *SRC4* may enable root systems to rapidly respond to these environmental signals and initiate corresponding adaptive responses.

Particularly noteworthy is that the availability of Ca^2+^ in the root environment provides ideal conditions for *SRC4* function. Soil solutions contain abundant Ca^2+^, and root cell membranes possess various Ca^2+^ transporters and channels that can precisely regulate intracellular Ca^2+^ concentrations [60]. Considering that *SRC4* protein can bind Ca^2+^ to alter its resistance intensity, the high Ca^2+^ environment in roots may provide favorable conditions for conformational activation and functional performance of *SRC4* protein. This spatial distribution matching suggests that tissue-specific expression of *SRC4* may be an important manifestation of its functional optimization.

High expression of *SRC4* in leaves is more directly related to its antiviral functions. Leaves are the primary target tissues for most plant virus infections, with viruses such as SMV being transmitted and infected through vector insects such as aphids feeding on leaves. High expression of *SRC4* in leaves ensures sufficient immune receptor proteins are available to recognize viral effector molecules and initiate defense responses in the early stages of viral infection. The rapid upregulation of *SRC4* expression within 2–5 h after SMV infection that we observed, combined with its high basal expression in leaves, forms an efficient viral defense system.

Although this study provides important insights into understanding *SRC4*’s unique regulatory mechanisms, some limitations remain that need to be addressed in future work. First, our research is mainly based on transcriptional level analysis, and we still lack in-depth understanding of *SRC4* protein post-translational modifications, subcellular localization dynamics, and interaction networks with other proteins. The molecular details of how Ca^2+^ binding precisely regulates *SRC4* protein conformational changes and functional domain activation, and whether *SRC4* has other ligand-dependent regulatory mechanisms, require further biochemical and structural biology studies for elucidation. Second, although we predicted the involvement of multiple transcriptional regulatory factors through promoter analysis, direct binding of these factors to the *SRC4* promoter and their interaction relationships have not been experimentally verified.

In terms of functional validation, current research is mainly conducted under laboratory-controlled conditions, and *SRC4*’s performance when facing complex and variable combinations of biotic and abiotic stresses in real field environments still requires large-scale field trials for verification. Additionally, our cross-species functional validation is mainly limited to *Arabidopsis* and tobacco model plants, and the functional conservation and application potential of *SRC4* in other important crops require broader validation. Epigenetic regulatory mechanisms, such as the roles of DNA methylation and histone modifications in *SRC4* expression regulation, are also important directions worthy of in-depth exploration.

### 3.1. Limitations

While this study provides valuable insights into *SRC4*’s transcriptional regulation, we acknowledge several areas that warrant further investigation. Our analysis primarily examined transcriptional-level regulation, while post-translational modifications of *SRC4* remain to be explored, particularly the precise molecular mechanisms of how Ca^2+^ binding affects protein conformation and activity. The promoter–transcription factor interactions were computationally predicted through bioinformatic analysis, though direct experimental validation of these binding events, especially for the SA-responsive and MYB binding sites, would strengthen our mechanistic understanding. Additionally, while our functional validation in heterologous model systems (*N. benthamiana* and *Arabidopsis*) provided valuable mechanistic insights, these systems may not fully capture the complex regulatory networks present in native soybean or the multi-stress interactions that occur under natural field conditions. Our tissue expression analysis was conducted at the organ level, and higher-resolution cellular localization studies would provide more detailed information about SRC4’s spatial organization and cell-type specificity within different tissues. Our tissue expression analysis was conducted at the organ level, which provides averaged expression data across multiple cell types with distinct mRNA profiles that may respond differently to external signals. Therefore, our organ-level expression data can only provide initial assumptions about SRC4 regulation rather than definitive conclusions about cellular-level mechanisms. Future studies with higher cellular resolution would be essential to achieve true cell-type specificity.

### 3.2. Future Directions

Building upon these foundations, several research directions would enhance our understanding of SRC4 function: biochemical characterization of SRC4 protein structure and Ca^2+^-induced conformational changes, experimental validation of predicted promoter-transcription factor interactions through chromatin immunoprecipitation or electrophoretic mobility shift assays, comprehensive field trials in soybean to evaluate dual stress resistance under natural multi-stress conditions, high-resolution cellular localization studies using advanced microscopic techniques, and investigation of SRC4’s broader role in integrating Ca^2+^-SA signaling networks. These studies will be essential for fully realizing SRC4’s potential in developing climate-resilient crops and deepening our understanding of its unique regulatory mechanisms.

## 4. Materials and Methods

### 4.1. Plant Materials and Growth Conditions

Soybean (*Glycine max*) cultivar Williams was used for gene expression analysis and promoter cloning. Seeds were surface-sterilized with 10% sodium hypochlorite for 10 min, rinsed five times with sterile water, and directly sown in a 1:1 mixture of nutrient soil and vermiculite at 25 °C under a 16 h light/8 h dark photoperiod. V3-stage soybean plants were used for SA treatment, Ca^2+^ treatment, and SMV inoculation experiments.

*Arabidopsis thaliana* ecotype Columbia-0 (Col-0) and transgenic tobacco (*Nicotiana benthamiana*) plants were grown in a controlled growth chamber at 22 °C under a 16 h light/8 h dark photoperiod with 60% relative humidity. *NahG*-overexpressing tobacco plants were kindly provided by Professor Jun Liu from China Agricultural University and maintained under the same conditions. The plant materials used in this study also included *Pro_SRC4_*::*GUS* transgenic *Arabidopsis*, Ox-*SRC4* At transgenic *Arabidopsis*, wild-type *N. benthamiana* 17Wt, and Ox-*SRC4* Nb transgenic tobacco plants, all maintained under the same growth conditions.

### 4.2. Transcriptome Data Analysis

Publicly available soybean RNA-seq datasets were downloaded from the NCBI Sequence Read Archive (SRA) database using SRA Toolkit v3.0.7. Raw sequencing data were converted to FASTQ format and subjected to quality control using FastQC v0.11.9. Expression quantification was performed using Kallisto v0.46.2 with the soybean reference transcriptome from Phytozome v13 (Glycine max Wm82.a4.v1). Transcript abundance was calculated as transcripts per million (TPM) values.

SMV infection-responsive expression data were obtained from three soybean cultivars: Williams (SRR5466715–SRR5466722), L29 (SRR10098755–SRR10098764), and 5031 (SRR6376246–SRR6376249). Differential expression analysis was performed comparing mock-inoculated controls with SMV-infected samples at 12 h post-inoculation (hpi).

Comprehensive expression analysis of SRC4 across different tissues, developmental stages, and stress conditions was conducted using the PlantRNAdb platform (http://ipf.sustech.edu.cn/pub/plantrna/, accessed on 11 May 2024). This database systematically integrates publicly available soybean RNA-seq data from major repositories including Gene Expression Omnibus (GEO), the Sequence Read Archive (SRA), the European Nucleotide Archive (ENA), and the DNA Data Bank of Japan (DDBJ). The 4085 soybean RNA-seq samples encompass diverse experimental conditions including various organs (roots, leaves, stems, seeds, flowers, pods), developmental stages (seedling, vegetative, reproductive, senescence), tissue types, and treatment conditions (biotic stress, abiotic stress, hormone treatments, and control conditions) with high coverage and data consistency. Expression levels were extracted as FPKM (fragments per kilobase of transcript per million mapped reads) values for comparative analysis.

### 4.3. Promoter Analysis and Vector Construction

Cis-acting regulatory elements were predicted for the 2000 bp genomic sequence upstream of the *SRC4* translation start site using PlantCARE software (http://bioinformatics.psb.ugent.be/webtools/plantcare/html/, Version: 1, accessed on 25 December 2023) [89]. The promoter sequence was retrieved from the soybean genome database (Phytozome v13, *Glycine max* Wm82.a4.v1, accessed on 25 December 2023) and analyzed for the presence of various regulatory motifs including hormone-responsive elements, stress-responsive elements, and tissue-specific elements.

Genomic DNA was isolated from young soybean (Williams cultivar) leaves using the CTAB (cetyltrimethylammonium bromide) method according to standard protocols. The *SRC4* promoter region (*Pro_SRC4_*, approximately 2000 bp upstream of ATG) was PCR amplified using high-fidelity DNA polymerase with specific primers containing Nimble Cloning (NC) universal adapter sequences (forward adapter: AGTGGTCTCTGTCCAGTCCT; reverse adapter: GGTCTCAGCAGACCACAAGT) [90].

The amplified *Pro_SRC4_* fragment was gel-purified and cloned into the pNC-121-Pro binary vector using the NC cloning system. The cloning reaction was performed in a total volume of 10 μL containing 5 μL Nimble Mix. The resulting *Pro_SRC4_*::*GUS* construct was transformed into *Escherichia coli* DH5α competent cells, and positive clones were verified by colony PCR and Sanger sequencing. Similarly, the *Pro_SRC4_*::*LUC* construct was generated by cloning the promoter fragment into the pNC-Green-LUC vector. For overexpression studies, the full-length *SRC4* coding sequence was cloned into the pNC-Cam3304-MCS35S vector downstream of the CaMV 35S promoter. All recombinant vectors were confirmed by restriction enzyme digestion and sequencing analysis before transformation into *Agrobacterium tumefaciens* strain GV3101.

### 4.4. Plant Transformation and Transgenic Line Generation

Transient expression assays were performed using 4–6 week-old *N. benthamiana* plants. *Agrobacterium* cultures containing the desired constructs were grown overnight at 28 °C, harvested by centrifugation, and resuspended in infiltration buffer (10 mM MES-KOH pH 5.6, 10 mM MgCl_2_, 100 μM acetosyringone) to an OD_600_ of 1.0. The bacterial suspension was infiltrated into the abaxial surface of fully expanded leaves using a needleless syringe. Plants were maintained under normal growth conditions and sampled at indicated time points.

The promoter-*GUS* recombinant vectors were infiltrated into *N. benthamiana* leaves for promoter activity analysis. For stress treatment assays, infiltrated plants were subjected to various treatments including SMV inoculation (OD_600_ = 1.0), SA treatment (0.5 mM), and Ca^2+^ treatment (10 mM CaCl_2_) at 24 h post-infiltration. Leaf discs (1 cm in diameter) were excised from the infiltrated areas at 48–72 h post-treatment and subjected to *GUS* histochemical staining using the *GUS* staining kit (G3060-100 mL Solarbio, Beijing, China) according to the manufacturer’s instructions. To validate the SA-dependency of *SRC4* promoter activation, the same constructs were also infiltrated into *NahG*-overexpressing tobacco plants.

*Arabidopsis* transformation was performed using the floral dip method [53]. *Agrobacterium* cells harboring the expression constructs were resuspended in 5% (*w*/*v*) sucrose solution containing 0.05% (*v*/*v*) Silwet L-77. Flowering *Arabidopsis* plants were inverted and dipped into the bacterial suspension for 30 s with gentle agitation. Treated plants were covered with plastic wrap and kept in darkness for 24 h, then returned to normal growth conditions. A second transformation was performed 7–10 days later to improve efficiency.

T_1_ seeds were surface-sterilized and plated on 1/2 MS medium containing appropriate selection agents: 50 mg/L kanamycin for *Pro_SRC4_*::*GUS* constructs (pNC-121-Pro vector), or 10 mg/L basta (glufosinate ammonium) for Ox-*SRC4* constructs (pNC-Cam3304 vector). Resistant seedlings were transferred to soil and grown to maturity. Homozygous T_3_ lines were identified by antibiotic/herbicide resistance segregation analysis and confirmed by genomic PCR using construct-specific primers. For tobacco transformation, *Agrobacterium* strains carrying the Ox-*SRC4* constructs were used to transform tobacco leaves following standard leaf disc transformation protocols.

### 4.5. Gene Expression Analysis

Total RNA was extracted from various plant tissues, including soybean root, stem, leaf, and flower tissues, as well as *Arabidopsis* leaves and whole plants, and the inoculation sites (punctured zones) of *N. benthamiana*, using TRIzol reagent (Invitrogen, Carlsbad, CA, USA) according to the manufacturer’s instructions. For comprehensive tissue-specific expression analysis, ten distinct tissues were collected from T_3_ *Pro_SRC4_::GUS Arabidopsis* plants at different developmental stages, including seedling roots (6-day-old seedlings), root tips (1–2 mm), cotyledons (6-day-old seedlings), rosette leaves (3-week-old plants), cauline leaves (4-week-old flowering plants), inflorescence stem (4-week-old flowering plants), flower buds (stage 9–12), open flowers (stage 13–14), pistils (hand-dissected from mature flowers), and young siliques (3–5 days post-fertilization). RNA quality and concentration were assessed using a NanoDrop spectrophotometer (Thermo Fisher Scientific, Waltham, MA, USA). First-strand cDNA was synthesized from 1 μg of total RNA using the HiScript III RT SuperMix for qPCR kit (Vazyme, Nanjing, China). RT-qPCR was performed using ChamQ Universal SYBR qPCR Master Mix (Vazyme, Q711) on a CFX96 Real-Time System (Bio-Rad, Hercules, CA, USA). Gene-specific primers were designed using Primer3Plus software (http://www.bioinformatics.nl/cgi-bin/primer3plus/primer3plus.cgi, version: 3.3.0, accessed on 1 September 2023). Soybean actin (*GmActin*), tobacco actin (*NbActin*), and *Arabidopsis* actin (*AtActin*) were used as internal reference genes for normalization. Relative expression levels were calculated using the 2^−ΔΔct^ method. All experiments included three biological replicates with three technical repeats each.

### 4.6. GUS Activity Analysis

For quantitative GUS analysis in transgenic *Arabidopsis*, total protein was extracted from leaf tissues using the Plant Protein Extraction Kit (BC3720, Solarbio **,** Beijing, China) according to the manufacturer’s instructions. Protein concentration was determined using the Bradford assay (SK1060, Coolaber, Beijing, China) and adjusted to 10 μg/mL. GUS activity was measured using a GUS quantitative detection kit (SL7161, Coolaber, Beijing, China) and a FilterMax F5 microplate reader (Molecular Devices, San Jose, CA, USA).

For promoter responsiveness assays in transgenic *Arabidopsis*, salicylate (SA) solution (0.5 mM) was applied as a whole-plant spray on T_3_ transgenic plants, and Ca^2+^ treatment was performed using 10 mM CaCl_2_ solution. *GUS* activity was determined by histochemical staining using the *GUS* staining kit (G3060, Solarbio, Beijing, China) according to the manufacturer’s instructions. Samples were collected at 48 h post-treatment for RT-qPCR analysis and at 72 h for protein activity quantification.

### 4.7. Stress Treatment Experiments

For salicylic acid (SA) treatment, V3-stage soybean plants or infiltrated tobacco leaves were sprayed with 0.5 mM SA solution. For Ca^2+^ treatment, V3-stage soybean plants were treated with 10 mM CaCl_2_ solution by foliar spray. Mock treatments used water. Samples were collected at 1, 2, 5, 12, and 24 h post-treatment for expression analysis.

SMV-N1 strain was mechanically inoculated onto V3-stage soybean leaves. Infected leaves carrying the virus strain were ground with 0.02 × PBS buffer and silica sand to prepare virus suspension, which was then applied by mechanical friction inoculation. Mock inoculations were performed using virus-free buffer. Leaf samples were collected at 1, 2, 5, 12, and 24 hpi for temporal expression analysis.

For temperature stress assays, plants were transferred to growth chambers set at 12 °C (cold stress) or 37 °C (heat stress), with mock controls maintained at 22 °C. Tobacco plants were treated for 7 days, while Arabidopsis plants were grown continuously for 25 days under these conditions. Plant phenotypes were photographed, and tissue samples were collected for gene expression analysis.

### 4.8. Statistical Analysis

All experiments were conducted with at least three biological replicates. Data are presented as mean ± SD deviation (SD). Statistical significance was determined using one-way ANOVA followed by Dunnett’s post-hoc test or Student’s *t*-test as appropriate. Significance levels were indicated as follows: * *p* < 0.05, and *** *p* < 0.001. *p*-values < 0.05 were considered statistically significant. Statistical analyses were performed using GraphPad Prism 8.0 software (GraphPad Software, San Diego, CA, USA).

## Figures and Tables

**Figure 1 plants-14-02820-f001:**
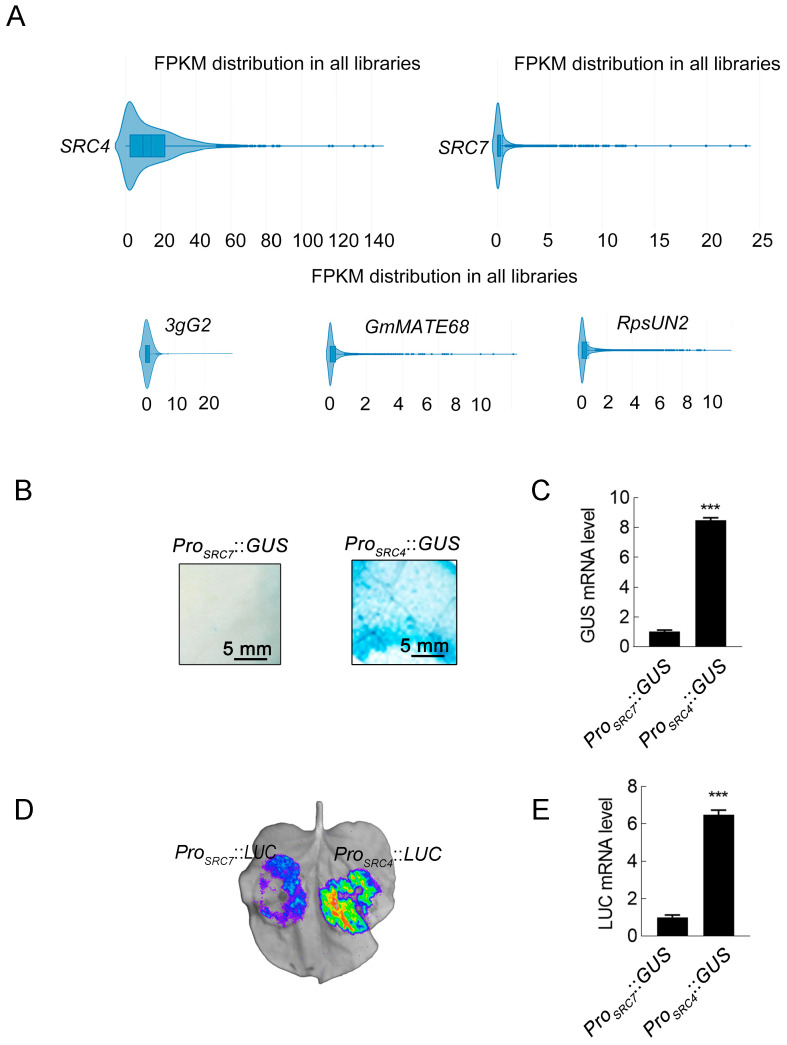
Expression analysis of *SRC4* promoter activities. (**A**) FPKM distribution analysis of *SRC4* and *SRC7* expression across soybean transcriptome databases. Expression data from 4085 soybean RNA-seq samples were retrieved from PlantRNAdb. Violin plots show the distribution of FPKM values for *SRC4* (left) and *SRC7* (right) across all libraries. (**B**) *GUS* staining assay for promoter activity comparison. Promoter sequences of *SRC7* and *SRC4* were cloned from soybean cultivar Williams and fused to *GUS* reporter gene in pNC-121-Pro binary vector. Constructs were transiently expressed in *N*. *benthamiana* leaves via *Agrobacterium*-mediated infiltration. *GUS* activity was visualized by X-Gluc staining at 48 h post-infiltration (hpi). Scale bar = 5 mm. (**C**) Quantitative RT-qPCR analysis of *GUS* mRNA levels in transiently expressed *N. benthamiana* leaves. Total RNA was extracted at 48 hpi and *GUS* expression levels were normalized to *N. benthamiana* actin. Data represent mean ± standard deviation (SD) of three biological replicates. (**D**) Luciferase complementation assay (LCA) for promoter activity. *Pro_SRC7_*::*LUC* and *Pro_SRC4_*::*LUC* constructs were infiltrated into *N. benthamiana* leaves and bioluminescence signals were captured at 48 hpi. Pseudocolor scale indicates signal intensity from low (blue) to high (red). (**E**) Quantitative RT-qPCR analysis of *LUC* mRNA levels. Expression levels were normalized to *N. benthamiana* actin gene. Data represent mean ± SD of three biological replicates, each with three technical repeats. Triple asterisks (***) indicate significant differences at *p* < 0.001 (Student’s *t*-test).

**Figure 2 plants-14-02820-f002:**
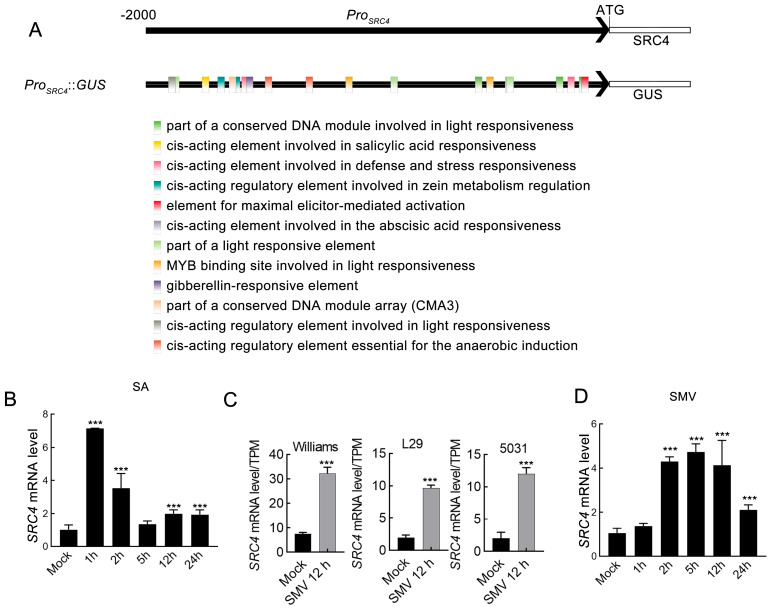
Transcriptional regulatory analysis of *SRC4*. (**A**) Schematic diagram of *Pro_SRC4_* promoter structure and *cis*-acting element prediction. The 2000 bp region upstream of *SRC4* coding sequence was obtained from soybean genome database, and *cis*-acting elements were predicted using PlantCARE database. Colored boxes indicate the positions and types of predicted regulatory elements in the promoter region. *Pro_SRC4_*::*GUS* fusion expression vector was constructed by cloning the promoter sequence into pNC-121-Pro binary vector. (**B**) Expression analysis of *SRC4* under salicylic acid (SA) treatment. Three-week-old soybean Williams seedlings were treated with foliar spray of 0.5 mM SA, and leaf samples were collected at 1, 2, 5, 12, and 24 h post-treatment (hpi). *SRC4* expression levels were detected by quantitative RT-qPCR and normalized to soybean actin gene (*GmActin*). (**C**) Expression response analysis of *SRC4* to SMV infection in different soybean cultivars. RNA-seq data from three soybean cultivars in NCBI database were analyzed: Williams (SRR5466715–SRR5466722), L29 (SRR10098755–SRR10098764), and 5031 (SRR6376246–SRR6376249). Expression levels are presented as transcripts per million (TPM) values, comparing expression differences between mock-inoculated plants and SMV-infected plants at 12 h post-infection (hpi). (**D**) Time-course analysis of *SRC4* expression after SMV-N1 infection. V3 stage soybean Williams leaves were rub-inoculated with SMV-N1, and samples were collected at 1, 2, 5, 12, and 24 h. *SRC4* expression levels were detected by quantitative RT-qPCR and normalized to *GmActin*. Data represent mean ± SD deviation (SD) of three biological replicates, each with three technical repeats. Statistical analysis was performed using one-way ANOVA post-hoc test. Triple asterisks (***) indicate extremely significant differences compared to mock control group (*p* < 0.001).

**Figure 3 plants-14-02820-f003:**
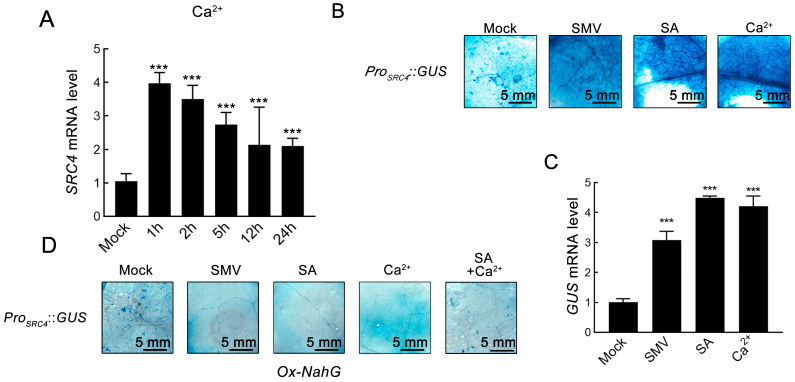
Analysis of Ca^2+^ signal-dependent regulatory mechanism of *SRC4* expression. (**A**) Time-course effects of exogenous Ca^2+^ treatment on *SRC4* expression. Three-week-old soybean Williams seedlings were treated with a foliar spray of 10 mM CaCl_2_ solution, and leaf samples were collected at 1, 2, 5, 12, and 24 hpi. *SRC4* relative expression levels were detected by quantitative RT-qPCR and normalized to soybean actin gene (*GmActin*). Data represent mean ± SD of three biological replicates, each with three technical repeats. Statistical analysis was performed using one-way ANOVA. Triple asterisks (***) indicate extremely significant differences compared to Mock control group (*p* < 0.001). (**B**) Response characteristics of *Pro_SRC4_* promoter to multiple signaling molecules. *Pro_SRC4_*::*GUS* fusion vectors were transiently transformed into tobacco (*N. benthamiana*) leaves via *Agrobacterium*-mediated method, with the following treatments: Mock (control group), SMV infection (OD_600_ = 1.0), SA treatment (0.5 mM salicylic acid foliar spray), and Ca^2+^ treatment (10 mM CaCl_2_ foliar spray). *GUS* activity was detected by X-Gluc staining 72 hpi. Scale bar = 5 mm. (**C**) Quantitative analysis of *GUS* expression under different treatment conditions. Quantitative RT-qPCR was used to detect relative expression levels of *GUS* mRNA in the above treatments, normalized to tobacco actin gene. Data represent mean ± SD deviation of three biological replicates. One-way ANOVA was used, with asterisks indicating significant differences compared to Mock group (*** *p* < 0.001). (**D**) Transient expression of *Pro_SRC4_*::*GUS* fusion vectors in transgenic tobacco overexpressing *NahG* gene (Ox-*NahG*), with the same treatments as wild-type: Mock, SMV infection, SA treatment, Ca^2+^ treatment, and combined SA and Ca^2+^ treatment (SA + Ca^2+^). *GUS* staining was detected 72 hpi. *NahG* gene encodes salicylate hydroxylase, which can degrade endogenous SA, used to verify SA-dependency of signal transduction. Scale bar = 5 mm.

**Figure 4 plants-14-02820-f004:**
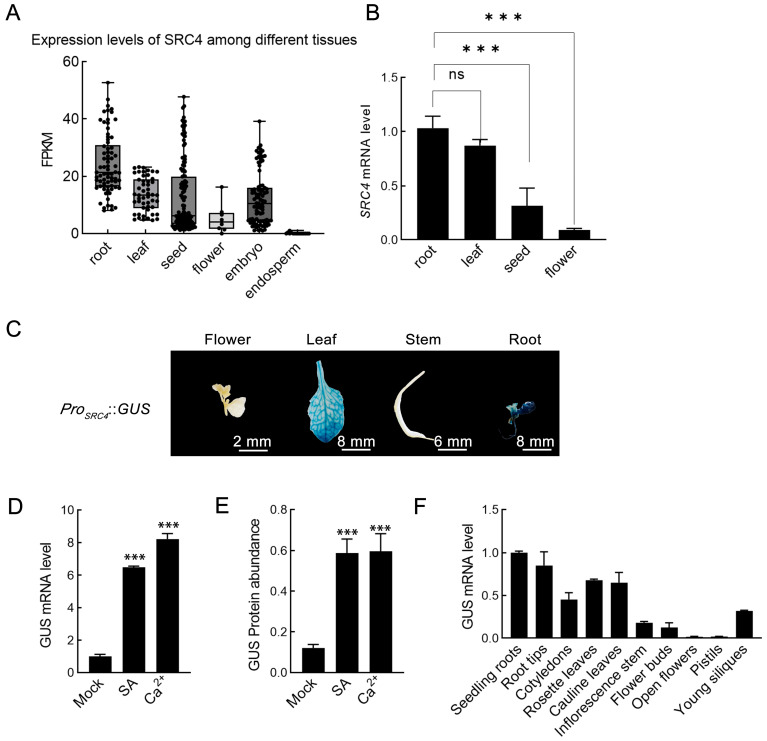
Tissue-specific expression pattern analysis and promoter functional verification of *SRC4*. (**A**) Expression level distribution of *SRC4* in different soybean tissues. Based on 4085 soybean transcriptome datasets from PlantRNAdb database, expression levels (FPKM) of *SRC4* in six tissues—root, leaf, seed, flower, embryo, and endosperm—were analyzed. Box plots show median, interquartile range, and outlier distribution of expression levels in each tissue. (**B**) Quantitative RT-qPCR verification of *SRC4* expression levels in different tissues of soybean Williams cultivar. *SRC4* relative expression in roots, leaves, seeds, and flowers was detected using soybean actin gene (*GmActin*) as internal reference. Data represent mean ± SD deviation of three biological replicates. Triple asterisks (***) indicate extremely significant differences compared to Mock group (*p* < 0.001). ns indicates no significant difference. (**C**) Tissue-specific expression analysis of *Pro_SRC4_*::*GUS* transgenic *Arabidopsis*. T_3_ transgenic *Arabidopsis* plants at flowering stage were stained with X-Gluc to show *GUS* activity distribution in different tissues. Scale bars: flower = 2 mm, leaf = 8 mm, stem = 6 mm, root = 8 mm. (**D**) Responsiveness detection of *Pro_SRC4_* to signaling molecules in transgenic *Arabidopsis*. T_3_ transgenic plants received SA treatment (0.5 mM SA foliar spray) and Ca^2+^ treatment (10 mM CaCl_2_ foliar spray), and *GUS* mRNA levels were detected by quantitative RT-qPCR 48 hpi, normalized to *Arabidopsis* actin gene. Data represent mean ± SD deviation of three biological replicates. Triple asterisks (***) indicate extremely significant differences compared to Mock group (*p* < 0.001). (**E**) Quantitative detection of *GUS* protein activity. Total leaf proteins from transgenic *Arabidopsis* were extracted 72 hpi, protein concentrations were adjusted to 10 μg/mL by Bradford method, and protein activity was measured using *GUS* quantitative detection kit. Data represent mean ± SD deviation of three biological replicates. Triple asterisks (***) indicate extremely significant differences compared to Mock group (*p* < 0.001) (**F**) Quantitative RT-qPCR analysis of *GUS* expression across different tissues in *Pro_SRC4_::GUS Arabidopsis*. T_3_ transgenic *Arabidopsis* plants were collected at different developmental stages, and *GUS* mRNA levels were detected in ten distinct tissues: seedling roots, root tips, cotyledons, rosette leaves, cauline leaves, inflorescence stem, flower buds, open flowers, pistils, and young siliques. *GUS* relative expression was normalized to *Arabidopsis* actin gene with seedling roots as calibrator. Data represent mean ± SD of three biological replicates.

**Figure 5 plants-14-02820-f005:**
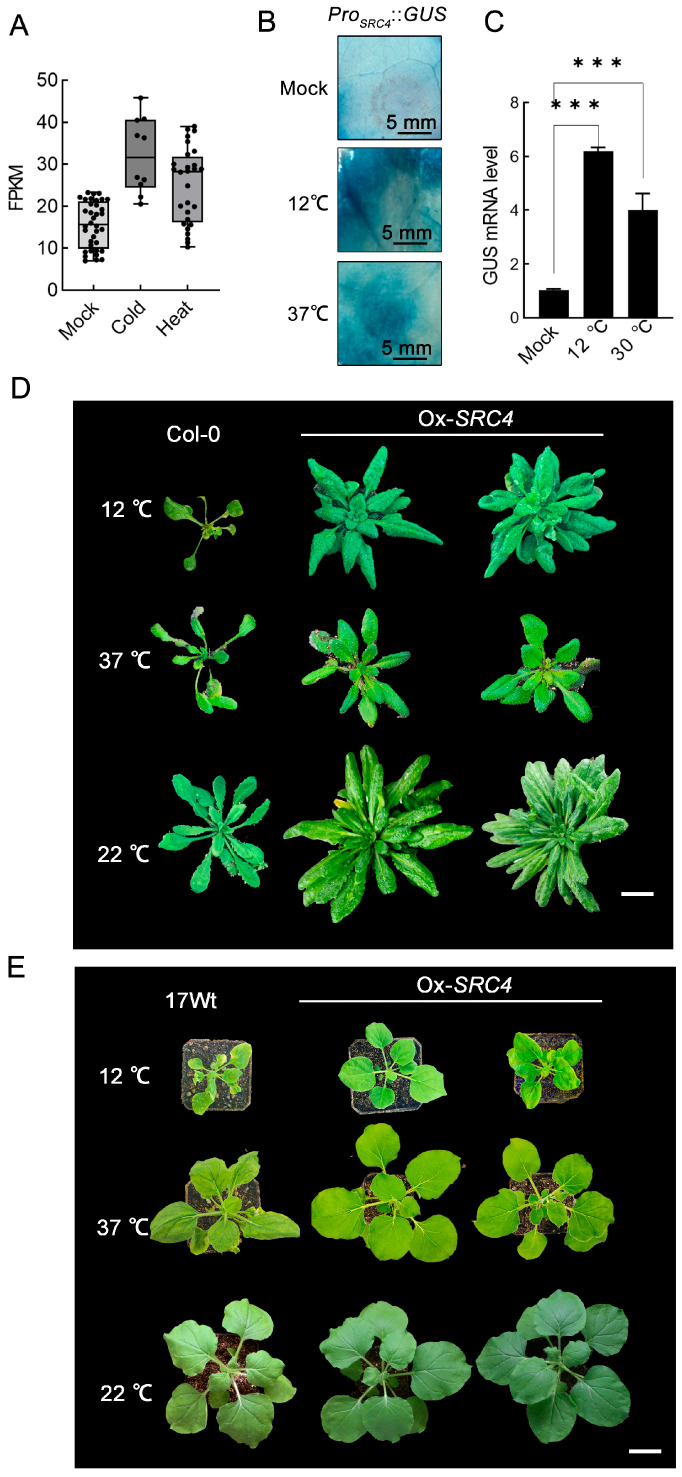
Response characteristics and functional analysis of *SRC4* to abiotic stress. (**A**) Expression level distribution of *SRC4* under different abiotic stress conditions in soybean. Based on 4085 soybean transcriptome datasets from PlantRNAdb database, expression levels (FPKM) of *SRC4* under normal conditions (Mock), cold stress (Cold), and heat stress (Heat) treatments were analyzed. Box plots show median, interquartile range, and outlier distribution of expression levels in each treatment group, with scatter points representing specific FPKM values in different samples. (**B**) Verification of *Pro_SRC4_* promoter responsiveness to temperature stress. *Pro_SRC4_*::*GUS* fusion vectors were transformed in a tobacco transient expression system, with 22 °C control (Mock), 12 °C low-temperature, and 37 °C high-temperature treatments, and *GUS* activity was detected by X-Gluc staining 7 days post-treatment. Scale bar = 5 mm. (**C**) Transcriptional response of *Pro_SRC4_* to temperature stress in transgenic *Arabidopsis*. T_3_ *Pro_SRC4_*::*GUS* transgenic *Arabidopsis* plants received 12 °C and 37 °C temperature treatments for 25 days, and *GUS* mRNA levels were detected by quantitative RT-qPCR, normalized to *Arabidopsis* actin gene. Data represent mean ± SD deviation of three biological replicates. Triple asterisks (***) indicate extremely significant differences compared to Mock group (*p* < 0.001). (**D**) Temperature stress tolerance analysis of Ox-*SRC4* transgenic *Arabidopsis*. T_3_ wild-type Col-0 and Ox-*SRC4* transgenic plants simultaneously received 12 °C low-temperature stress, 37 °C high-temperature stress, and 22 °C control treatments for 25 days, and plant phenotypes were photographed. Scale bar = 1 cm. (**E**) Temperature stress tolerance analysis of Ox-*SRC4* transgenic tobacco. T_3_ wild-type 17Wt and three independent Ox-*SRC4* transgenic tobacco lines received 12 °C low-temperature stress, 37 °C high-temperature stress, and 22 °C control treatments for 21 days, and plant phenotypes were photographed. Scale bar = 2 cm.

**Table 1 plants-14-02820-t001:** Major regulatory elements and positions of the *SRC4* promoter.

Element Position (bp)	Function Description	Element Position (bp)	Function Description
−1906~−1876	Light-responsive element	−1144~−1114	MYB binding site, involved in light response
−1905~−1875	Abscisic acid-responsive element	−950~−920	Light-responsive element
−1898~−1868	Light-responsive element	−586~−556	Light-responsive element
−1761~−1731	Salicylic acid-responsive element	−537~−507	MYB binding site, involved in light response
−1694~−1664	Zein metabolism regulatory element	−457~−427	Light-responsive element
−1645~−1615	Conserved DNA module (CMA3)	−450~−420	Light-responsive element
−1627~−1597	Zein metabolism regulatory element	−237~−207	Conserved DNA module, involved in light response
−1591~−1561	Defense and stress-responsive element	−188~−158	Defense and stress-responsive element
−1571~−1541	Gibberellin-responsive element	−139~−109	Light-responsive element
−1490~−1460	Anaerobic induction essential element	−129~−99	Inducible activation element
−1314~−1284	Anaerobic induction essential element	−1906~−1876	Light-responsive element

## Data Availability

The data presented in this study are available in the following public repositories: 1. Soybean transcriptome data: PlantRNAdb (Plant Public RNA-seq Database) at http://ipf.sustech.edu.cn/pub/plantrna/ (accessed on 15 November 2023). A total of 4085 soybean RNA-seq datasets were analyzed for SRC4 expression profiling. 2. SMV infection transcriptome data: NCBI Sequence Read Archive (SRA) at https://www.ncbi.nlm.nih.gov/sra with the following accession numbers: Williams cultivar: SRR5466715–SRR5466722, L29 cultivar: SRR10098755–SRR10098764, 5031 cultivar: SRR6376246–SRR6376249. 3. Soybean genome sequence: Phytozome v13 database at https://phytozome-next.jgi.doe.gov/ (accessed on 25 December 2023), Glycine max Wm82.a4.v1 assembly. 4. Promoter cis-regulatory element prediction: PlantCARE database at http://bioinformatics.psb.ugent.be/webtools/plantcare/html/ (Version 1, accessed on 25 December 2023). All other data generated during this study, including experimental measurements and transgenic plant materials, are included in this published article and its Appendix A.

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
