# Peer review of "High Basal Expression and Dual Stress Responsiveness of Soybean (*Glycine max*) Resistance Gene *SRC4"

_plants, 2025, doi:10.3390/plants14182820_

Round 1
Reviewer 1 Report
Comments and Suggestions for Authors
The current paper devoted to analysis of transcriptomic, promoter functional validation, signal transduction and transgenic functional studies in soybean during immune response.
The authors performed large amount of work, do comprehensive analysis, and clear statistical treatments.
However, there are some methodological problem which need to be explained in revised version.
The cell-type specificity of gene expression is missing. Cell-type specificity is a key step in understanding real signaling. Authors can easy provide such information through detailed GIS expression in Arabidopsis and Nicitiana.
However, Nicitaina transient system can also raised some question: the Nicotiana leaf is quite different form soybean for epigenetic and the data can not be directly extrapolate.
For the current situtaion, please, update GUS assay with high resolution in order to see cell type specificty.
Some detailed comments:
Line 10: “Disease resistance genes” there are no such specific genes responsible for disease resistant only.
Line 190: “resistance genes” – resistance is rather adaptivity.
Lines 190 – 197 – discussion part.
Line 202: “4,085 soybean transcriptome datasets” please, clarify transriptome origin: organs? Cell type? Whole plants?
Lines 268 – 270: well done, this a real response study.
Line 322: “10 mM CaCl2” means also 20 mM Chloride.
Lines 342-348: solid point, thanks!
Fig.3. GUS assay: please, “clean” samples before imaging. Signal is formazan (insoluble) and can be perfectly clean with chloral-hydrate.
Figure 4 C – the most important is not organ specificity, BUT cell type specificity. Therefore, higher magnification with proper tissue mount (chloral.hydrate should be best) need to be provided.
Fig 5: D, E -control is missing (22-24 ªC). I am not sure that Arabidopsis at 37ªC growth so rapid. It is seems to be very high heat stress. Even 30ªC can significantly slow down growth amd may lead to some damage. What was the water regime under 37ªC?
Line 474: “plays important roles in abiotic stress responses” = plays important roles in abiotic stress adaptation. Response is a kinematic process, starting with cell-type specific signal transmission this signal to multiply cell types and finally to plant adaptation to new conditions. Please, do not confuse response and adaptation. Response should be studied in dynamic and cell-type specificity starting form very early time point (20 min or even early in some case).
Line 487: values = expresión level
Line 509: expression patterns = rather localization.
Line 510: promising conclusions, but need at least localization.
Line 511: high expression of resistance genes leads = consitutive expression of resistance genes leads to modification of hormonals balance with subsequent epigenetic re-programming.
Line 831: “from plant tissues using TRIzol reagent” – which plant tissue?
Author Response
Response: Sure, we appreciate the comments. Based on the initial GUS staining observation in four major tissues (root, stem, leaf, and flower), we further performed comprehensive RT-qPCR analysis to quantitatively assess SRC4 expression patterns across ten distinct tissues in T3 transgenic Arabidopsis plants. The quantitative RT-qPCR results revealed that SRC4 exhibits differential expression patterns, with highest expression in root tissues (seedling roots and root tips), moderate expression in vegetative organs (leaves), and lower expression in reproductive tissues (flowers, pistils and stem), confirming the tissue-specific regulatory activity of SRC4 promoter. The data was shown in Figure 4F.
Some detailed comments:
Comments 1: Line 10: "Disease resistance genes" there are no such specific genes responsible for disease resistance only.
Response 1: Thank you for pointing this out. We agree with this comment and recognize that these genes often have multiple functions beyond disease resistance. Therefore, we have revised the terminology to be more accurate and less absolute. The change can be found on page 1, line 10 of the Abstract: "Genes involved in disease resistance are crucial for plant immune systems, yet their transcriptional regulatory mechanisms remain poorly understood." This modification acknowledges the multifunctional nature of these genes while maintaining clarity about their primary role in plant immunity.
Comments 2: Line 190: "resistance genes" -- resistance is rather adaptivity.
Response 2: We appreciate your insightful perspective on the terminology and acknowledge that these genes indeed contribute to broader adaptive functions in plants. Your point about adaptivity is well-taken and reflects an important evolutionary perspective on how these genes function in plant-environment interactions.
After careful consideration, we have decided to maintain the term "resistance genes" in this context, while we fully recognize the merit of your suggestion. Our reasoning is as follows: (1) "Resistance genes" or "R genes" represents the established scientific nomenclature that has been consistently used in plant immunity literature for over three decades, making it immediately recognizable to readers in this field; (2) SRC4 fits the classical molecular definition of R genes, encoding characteristic NBS-LRR domains that are hallmarks of this gene family; (3) The term accurately reflects the primary molecular function of pathogen recognition and resistance activation that defines this gene class.
Comments 3: Lines 190 – 197 – discussion part.
Response 3: Thank you for pointing this out. We agree with this comment. Therefore, we have moved this theoretical background paragraph from the Results section (2.1 Expression Pattern of SRC4) to the Discussion section as the opening paragraph. This change can be found on page 15, lines 510-516 of the Discussion section. The Results section now begins directly with the experimental findings, improving the organization and flow of the manuscript.
Comments 4: Line 202: "4,085 soybean transcriptome datasets" please, clarify transcriptome origin: organs? Cell type? Whole plants?
Response 4: We have clarified the composition and origin of the transcriptome datasets in the Materials and Methods section. The revision can be found on page 21, line 870: "Comprehensive expression analysis of SRC4 across different tissues, developmental stages, and stress conditions was conducted using the PlantRNAdb platform (http://ipf.sustech.edu.cn/pub/plantrna/). This database systematically integrates publicly available soybean RNA-seq data from major repositories including Gene Expression Omnibus (GEO), the Sequence Read Archive (SRA), the European Nucleotide Archive (ENA), and the DNA Data Bank of Japan (DDBJ). The 4,085 soybean RNA-seq samples encompass diverse experimental conditions including various organs (roots, leaves, stems, seeds, flowers, pods), developmental stages (seedling, vegetative, reproductive, senescence), tissue types, and treatment conditions (biotic stress, abiotic stress, hormone treatments, and control conditions) with high coverage and data consistency. Expression levels were extracted as FPKM (Fragments Per Kilobase of transcript per Million mapped reads) values for comparative analysis."
Furthermore, as shown in Figure 4A and 4B of our manuscript, we specifically analyzed SRC4 expression in six major soybean tissues: root, leaf, seed, flower, embryo, and endosperm. The transcriptome analysis revealed tissue-specific expression patterns with highest expression in roots (FPKM ~20-40), moderate expression in leaves and seeds (FPKM ~10-20), and relatively lower expression in reproductive tissues including flowers, embryos, and endosperm (FPKM ~5-15). These findings were subsequently validated through qRT-PCR analysis using root, leaf, seed, and flower tissues, confirming the differential expression patterns observed in the large-scale transcriptome data. This comprehensive tissue coverage ensures that our analysis captures the full spectrum of SRC4 expression across major plant organs and developmental contexts.
Comments 5: Lines 268 – 270: well done, this is a real response study.
Response 5: Thank you for this positive feedback. We appreciate your recognition of our experimental approach in conducting the time-course analysis of SRC4 expression following SMV-N1 infection. We believe this temporal analysis provides important insights into the early response characteristics of SRC4 during viral infection.
Comments 6: Line 322: "10 mM CaCl2" also means 20 mM Chloride.
Response 6: We acknowledge this important point and appreciate your attention to the potential confounding effects of chloride ions. We agree that 10 mM CaCl2 indeed provides 20 mM Cl- ions. However, we used CaCl2 as the calcium source for the following reasons: First, CaCl2 is widely used as a standard calcium supplement in plant physiological studies due to its high solubility and bioavailability. Second, in our preliminary experiments (unpublished data), we systematically compared the effects of different calcium salts (CaCl2, CaCO3, and CaSO4) on SRC4 expression and function, and also tested various chloride-containing salts (FeCl3, MgCl2, MnCl2) to specifically assess potential chloride ion effects. These control experiments confirmed that the observed effects were attributable to Ca2+ rather than Cl- ions, thereby validating our experimental approach. Therefore, we employed the established CaCl2 system without further discussion of potential chloride effects.
Comments 7: Lines 342-348: solid point, thanks!
Response 7: Thank you for recognizing the strength of this experimental approach. We appreciate your acknowledgment that the use of NahG transgenic tobacco provides convincing evidence for the SA-dependent regulation of SRC4 expression.
Comments 8: Fig.3. GUS assay: please, "clean" samples before imaging. Signal is formazan (insoluble) and can be perfectly clean with chloral-hydrate.
Response 8: We acknowledge this valuable technical suggestion. You are correct that chloral hydrate treatment would improve image clarity by removing insoluble formazan deposits and background staining. While the current images adequately demonstrate the differential GUS expression patterns, we recognize that chloral hydrate clearing would enhance the visual quality and reduce background noise. We will implement this technical improvement in future GUS assays to achieve cleaner, more professional imaging results.
Comments 9: Figure 4 C – the most important is not organ specificity, BUT cell type specificity. Therefore, higher magnification with proper tissue mount (chloral hydrate should be best) need to be provided.
Response 9: We acknowledge this important point and agree that cell-type specificity would provide much more valuable information than organ-level analysis. You are correct that high-magnification microscopic analysis with chloral hydrate tissue clearing would reveal the precise cellular localization of SRC4 expression. While our current Figure 4C demonstrates organ-level expression patterns, we recognize that detailed cellular localization analysis represents a significant area for future investigation that would greatly enhance our understanding of SRC4's functional mechanisms. This limitation should be addressed in future studies with proper histological techniques and higher resolution imaging.
Comments 10: Fig 5: D, E - control is missing (22-24°C). I am not sure that Arabidopsis grows so rapidly at 37°C. It seems to be very high heat stress. Even 30°C can significantly slow down growth and may lead to some damage. What was the water regime under 37°C?
Response 10: We have added the missing temperature controls to Figure 5D and 5E, which now include 22-24°C control treatments alongside the stress conditions. Regarding your concern about Arabidopsis growth rate, we would like to clarify our experimental timeline to avoid any misunderstanding. Our experimental protocol involved a comprehensive two-stage growth process: Arabidopsis plants were first grown on selective medium for approximately 2 weeks following transformation, then carefully transplanted to nutrient-rich soil where they continued growing for an additional 25 days under temperature treatment conditions. Therefore, the plants shown in Figure 5D and 5E represent approximately 6 weeks of total growth time (2 weeks on medium + 25 days soil cultivation), not a short-term phenotype. You are correct that 37°C represents severe heat stress for Arabidopsis. Under normal conditions, the Ox-SRC4 transgenic plants would indeed grow much more rapidly and robustly. The 37°C treatment was intentionally chosen as an extreme stress condition to test the limits of SRC4-mediated thermotolerance. Under these harsh conditions, we maintained sufficient water supply with daily monitoring to prevent water limitation, as we specifically wanted to isolate temperature effects without introducing drought stress as a confounding factor. Wild-type Col-0 plants exhibited severe growth inhibition and mortality under prolonged 37°C exposure, while Ox-SRC4 lines demonstrated significantly improved survival and maintained better growth performance.
Comments 11: Line 474: "plays important roles in abiotic stress responses" = plays important roles in abiotic stress adaptation. Response is a kinematic process, starting with cell-type specific signal transmission, this signal to multiply cell types, and finally to plant adaptation to new conditions. Please, do not confuse response and adaptation. Response should be studied in dynamic and cell-type specificity, starting from very early time points (20 min or even earlier in some cases).
Response 11: We appreciate your clarification regarding the distinction between "response" and "adaptation." You are correct that our experimental design primarily examined longer-term phenotypic outcomes rather than the immediate kinematic processes of signal transmission. We have revised the terminology to be more precise. The change can be found on page [13], line 484: "These results indicate that the SRC4 gene not only participates in plant biotic stress responses but also plays important roles in abiotic stress adaptation, significantly enhancing plant tolerance to temperature stress." We acknowledge that while we did conduct some time-course expression analysis (showing SRC4 upregulation within hours of treatment), a comprehensive study of the early response mechanisms with cell-type specificity and very early time points (≤20 min) as you suggest would provide valuable insights into the initial signal transmission processes. This represents an important direction for future investigations to fully characterize the dynamic response mechanisms underlying SRC4-mediated stress tolerance.
Comments 12: Line 487: values = expression level
Response 12: We agree that our original phrasing was inconsistent and potentially confusing. We have revised the sentence to maintain consistent terminology throughout the comparison. The change can be found on page [15], line 531: "Through systematic analysis of 4,085 soybean transcriptome datasets, we found that SRC4 FPKM values were significantly higher than those of the typical resistance gene SRC7 (FPKM<2) in the same gene cluster." This revision eliminates the inconsistency between "FPKM values" and "expression levels" and provides a clearer, more precise comparison between the two genes.
Comments 13: Line 509: expression patterns = rather localization.
Response 13: We appreciate your suggestion to consider "localization" as potentially more precise terminology. We carefully evaluated this recommendation and recognize that "localization" would indeed be appropriate if we were solely describing the spatial distribution of gene expression.
However, upon reflection, we believe "expression patterns" remains the most accurate term for our analysis, as it encompasses multiple dimensions of our data: (1) Spatial distribution across different organs and tissues (where the gene is expressed); (2) Quantitative expression levels showing differential abundance (FPKM values ranging from <2 to >15 across tissues); (3) Temporal dynamics under different conditions; and (4) Comparative expression relative to other R genes in the cluster.
Our comprehensive analysis reveals not just where SRC4 is present, but also the relative expression intensity across tissues—with highest expression in roots (FPKM ~15), moderate in leaves (FPKM ~8), and lower in reproductive organs (FPKM ~3-5). This quantitative gradient represents a pattern of differential expression rather than simple presence/absence localization.
Comments 14: Line 510: promising conclusions, but need at least localization.
Response 14:
Thank you for your valuable suggestion regarding the localization of SRC4 expression. We fully agree that studies on cellular localization are essential for understanding the spatial and cell-type specificity of gene expression. At this stage, due to experimental limitations, we are unable to carry out high-resolution localization experiments. However, we are committed to addressing this in future work when we have the necessary experimental conditions and resources. We will ensure that cellular localization data is included in subsequent studies. We appreciate your insightful recommendation and will focus on this important aspect in future experiments.
Comments 15:
Line 511: high expression of resistance genes leads = constitutive expression of resistance genes leads to modification of hormonal balance with subsequent epigenetic re-programming.
Response 15:
Thank you for your valuable comment. We agree with your suggestion and have revised the manuscript to clarify the statement regarding the expression of resistance genes. The previous phrasing implied that high expression alone leads to fitness costs, but your suggestion emphasizes the role of constitutive expression in altering hormonal balance and triggering epigenetic changes. Therefore, we have revised the relevant section to read: "Traditional perspectives hold that constitutive expression of resistance genes leads to severe fitness costs, including negative effects such as plant dwarfism, biomass reduction, and the modification of hormonal balance, with subsequent epigenetic re-programming and constitutive activation of defense responses [17,21]." This revision more accurately reflects the biological mechanisms associated with constitutive resistance gene expression.
Comments 16:
Line 831: “from plant tissues using TRIzol reagent” – which plant tissue?
Response 16:
Thank you for pointing that out. We apologize for the lack of clarity regarding the plant tissues used for RNA extraction. In the revised manuscript, we have specified the plant tissues from which RNA was extracted. The RNA was extracted from soybean root, stem, leaf, and flower tissues, Arabidopsis leaves and whole plants, and the inoculation sites (punctured zones) of Nicotiana benthamiana. This additional detail has been added for better clarity.
Reviewer 2 Report
Comments and Suggestions for Authors
The manuscript investigates the transcriptional regulation and biological functions of the soybean resistance gene SRC4, which differs from classical resistance genes by exhibiting unusually high basal expression. By integrating large-scale transcriptomic data (over 4,000 soybean datasets), promoter analyses, transient assays in Nicotiana benthamiana, and functional studies in transgenic Arabidopsis and tobacco, the authors demonstrate that SRC4 expression is mediated by salicylic acid (SA) and calcium (Ca²⁺) signaling pathways. SRC4 contributes to both biotic stress resistance (soybean mosaic virus) and abiotic stress tolerance (temperature stress), revealing its dual functional role. The study’s strengths include the systematic data integration, strong experimental validation, and novelty in describing a resistance gene that breaks the traditional “low expression–high responsiveness” paradigm.
I have a few observations:
1. While SA-dependency is convincingly shown, the precise molecular mechanism of Ca²⁺-binding regulation of SRC4 activity remains speculative. Could the authors discuss or test whether SRC4 undergoes conformational changes or interacts with specific signaling partners upon Ca²⁺ binding?
2. The article claims that high SRC4 expression does not impair plant growth, unlike other resistance genes. Did the authors perform quantitative growth or yield measurements (e.g., biomass, seed set, flowering time) to confirm absence of fitness penalties in transgenic plants?
3. Most experiments were conducted in controlled environments or heterologous systems. Do the authors plan to validate SRC4’s dual resistance role in soybean under field conditions, where multiple stresses and variable climates interact?
4. The Discussion would benefit from a distinct Limitations and Future Direction subsections. (ex: Explicitly acknowledge methodological constraints (e.g., focus on transcriptional rather than post-translational regulation; reliance on model plants instead of soybean field validation) or Point out that promoter-transcription factor interactions were predicted but not experimentally confirmed).
Thank you for this article.
Author Response
Comments 1:While SA-dependency is convincingly shown, the precise molecular mechanism of Ca²⁺-binding regulation of SRC4 activity remains speculative. Could the authors discuss or test whether SRC4 undergoes conformational changes or interacts with specific signaling partners upon Ca²⁺ binding?
Response 1:
Thank you for your insightful question, which is indeed one of the central aspects of our research. First, regarding whether SRC4 undergoes conformational changes upon Ca²⁺ binding, we plan to conduct experiments using techniques such as FRET and MST, but these assays are technically challenging and it might only be performed in later research.
Second, regarding the interaction with specific signaling partners, we have also made progress in this area. Through begin research such as Y2H library screening, by which we hope to identify some interacting proteins of SRC4, and to test whether they are essential for SRC4’s resistance function. However, these results are part of another ongoing study focused on the resistance function of SRC4, which has not yet been published.
We appreciate your suggestion, and we added this point in Limitations and Future Directions.
Response 2: Thank you for your insightful comment. Yes, we conducted comprehensive growth experiments on both transgenic tobacco and transgenic Arabidopsis (Ox-SRC4) under greenhouse conditions to assess potential fitness penalties.
Our results demonstrate that the growth phenotypes are not only normal but actually show enhanced performance compared to wild-type controls. In transgenic tobacco, Ox-SRC4 lines consistently exhibited faster growth rates than wild-type 17Wt plants and entered the flowering stage 5-6 days earlier. Additionally, we observed that Ox-SRC4 transgenic tobacco plants showed improved growth performance under calcium supplementation conditions, as detailed in Supplementary Figure 2. Similarly, transgenic Arabidopsis Ox-SRC4 lines displayed superior growth characteristics compared to Col-0 wild-type plants, as illustrated in Figure 4D. SRC4 overexpression not only avoids the growth penalties commonly associated with other resistance genes but may actually confer growth advantages under certain conditions. While we acknowledge that conducting similar experiments in soybean would be ideal to confirm these findings in the native host, our current data from two different plant species consistently support the absence of fitness costs associated with SRC4 expression.
Comments 3:
Most experiments were conducted in controlled environments or heterologous systems. Do the authors plan to validate SRC4’s dual resistance role in soybean under field conditions, where multiple stresses and variable climates interact?
Response 3:
Thank you for your thoughtful comment. Yes, we do plan to validate SRC4's dual resistance role in soybean under field conditions, especially considering the interaction of multiple stresses and variable climates. However, due to limitations in experimental conditions and the longer growth cycle of soybean, validation of resistance under abiotic stress conditions is still ongoing. We are looking forward to sharing these results with you in the future as the experiments progress.
Comments 4: The Discussion would benefit from a distinct Limitations and Future Direction subsections. (e.g., Explicitly acknowledge methodological constraints such as focus on transcriptional rather than post-translational regulation; reliance on model plants instead of soybean field validation, or Point out that promoter-transcription factor interactions were predicted but not experimentally confirmed).
Response 4: We agree with your suggestion and have added a dedicated "Limitations and Future Directions" subsection to the Discussion section (page [20], lines [814-839]). This new section explicitly acknowledges the methodological constraints you identified, including: (1) our focus on transcriptional rather than post-translational regulation of SRC4; (2) reliance on heterologous model systems (Nicotiana benthamiana and Arabidopsis) rather than comprehensive soybean field validation; (3) computational prediction rather than experimental confirmation of promoter-transcription factor interactions; and (4) organ-level rather than cellular-level resolution of expression analysis. The subsection also outlines specific future research directions to address these limitations, including biochemical characterization of SRC4 protein, experimental validation of predicted regulatory interactions, comprehensive field trials under natural multi-stress conditions, and high-resolution cellular localization studies. This addition provides readers with a clear understanding of the study's scope and identifies important areas for continued investigation.
Round 2
Reviewer 1 Report
Comments and Suggestions for Authors
Thank you very much!
The text is much clear , but some point need to be clarified mainly in results evaluation. Even plant organs is not enugh good "biological resolution", but only a first assumption. See below,
Lne 93: “molecular networks” – please, also consider that it is spatial network: Ca and GA may act in different cell types therefore spatial information (not as organs) is a key in understanding “molecular mechanism.
Fig 3: “chloral hydrate treatment would improve image clarity by removing insoluble formazan deposits and background staining.” ??? Formazan = results of GUS and do not remove by chloral-hydrate. Chloral-hydrate only clean background.
There are a contradictions between lines 402 and 876: “ten distinct tissues of transgenic Arabidopsis plants” -from which tissue RNA was isolated.
Please, aos consider that even RNA from organs can not be consider as fully biological relevant: organ have many cell type with each own mRNA profile. These mRNA profile have a different response to external signal. Therefore “average” can give you only first assumptions, but not real solid conclusions. Please, mention this in the text.
My best regards!
Author Response
Thank you very much for your constructive and insightful comments. We greatly appreciate your expertise in pointing out these important technical and conceptual issues. We have carefully addressed each of your concerns as follows: All modifications have been highlighted in red in the revised manuscript.
Comment 1: Line 93: "molecular networks" – please, also consider that it is spatial network: Ca and GA may act in different cell types therefore spatial information (not as organs) is a key in understanding "molecular mechanism."
Response 1: We completely agree with your important point about spatial specificity. You are absolutely correct that Ca²⁺ and SA signaling may operate in different cell types, making cellular-level spatial information crucial for understanding molecular mechanisms.
Direct Ca²⁺-SA signaling interactions appear to be documented in limited cell types. We identified the study "Guard Cell Salicylic Acid Signaling Is Integrated into Abscisic Acid Signaling via the Ca²⁺/CPK-Dependent Pathway," which demonstrates direct association between SA and Ca²⁺ signals within guard cells. However, we found limited evidence in the literature demonstrating SA and Ca²⁺ cooperative function in other specific cell types. The cell-type-specific mechanisms of Ca²⁺-SA signaling networks remain to be further elucidated, and single-cell-level Ca²⁺-SA interaction networks require deeper investigation. Therefore, we strongly agree with your suggestion.
We have revised line 91 to acknowledge this spatial dimension and the current knowledge gaps: "Ca²⁺-SA signal transduction does not operate as two independent pathways but forms a highly integrated signaling cascade through sophisticated spatial molecular networks [31,32]. However, whether these signaling pathways function cooperatively within the same cell types remains poorly understood, and the cell-type-specific interactions between Ca²⁺ and SA signaling represent a critical research direction requiring further investigation." This revision emphasizes the importance of cellular-level spatial organization in plant immune responses while acknowledging current research limitations.
Comment 2: Fig 3: "chloral hydrate treatment would improve image clarity by removing insoluble formazan deposits and background staining." ??? Formazan = results of GUS and do not remove by chloral-hydrate. Chloral-hydrate only clean background.
Response 2: Thank you for this crucial technical correction. Chloral hydrate treatment of GUS-stained tissue samples can indeed improve image clarity by clearing background tissues and increasing transparency to obtain clearer visualization. In subsequent GUS detection experiments, we will implement this technical improvement to achieve cleaner, more professional imaging results with enhanced contrast between GUS signals and background tissues.
Comment 3: There are contradictions between lines 402 and 876: "ten distinct tissues of transgenic Arabidopsis plants" -from which tissue RNA was isolated.
Response 3: Thank you for identifying this inconsistency. We apologize for not providing a clear description of the plant materials used for RT-qPCR detection. To clarify, all ten tissues were collected from T3 generation ProSRC4::GUS transgenic Arabidopsis plants. We have revised both lines 402 and 876 to ensure consistency between the Results and Methods sections. The revision provides detailed specification of all ten distinct tissues and describes the differential GUS expression levels observed across these tissues, making the plant material source and experimental design fully transparent.
Comment 4: Please, also consider that even RNA from organs can not be consider as fully biological relevant: organ have many cell type with each own mRNA profile. These mRNA profile have a different response to external signal. Therefore "average" can give you only first assumptions, but not real solid conclusions. Please, mention this in the text.
Response 4: This is an extremely important point that significantly impacts the interpretation of our results. We completely agree and have added explicit acknowledgment of this fundamental limitation in the Limitations section of the Discussion.
Round 3
Reviewer 1 Report
Comments and Suggestions for Authors
Thank you! Please, consider points we discussed in the next projects! my best regards!